# Buffering of transcription rate by mRNA half-life is a conserved feature of Rett syndrome models

**Deivid C. Rodrigues** [1,6], **Marat Mufteev**[1,2,6], **Kyoko E. Yuki**[3], **Ashrut Narula** [2,4], **Wei Wei**[1], **Alina Piekna**[1], **Jiajie Liu**[1], **Peter Pasceri**[1], **Olivia S. Rissland** [4,5], **Michael D. Wilson** [2,3] & **James Ellis** [1,2] ✉

Transcriptional changes in Rett syndrome (RTT) are assumed to directly correlate with steady-state mRNA levels, but limited evidence in mice suggests that changes in transcription can be compensated by post-transcriptional regulation. We measure transcription rate and mRNA half-life changes in RTT patient neurons using RATEseq, and re-interpret nuclear and whole-cell RNAseq from *Mecp2* mice. Genes are dysregulated by changing transcription rate or half-life and are buffered when both change. We utilized classifier models to predict the direction of transcription rate changes and find that combined frequencies of three dinucleotides are better predictors than CA and CG. MicroRNA and RNA-binding Protein (RBP) motifs are enriched in 3′UTRs of genes with half-life changes. Nuclear RBP motifs are enriched on buffered genes with increased transcription rate. We identify post-transcriptional mechanisms in humans and mice that alter half-life or buffer transcription rate changes when a transcriptional modulator gene is mutated in a neurodevelopmental disorder.

Rett syndrome (RTT) is a neurodevelopmental disorder in girls caused by damaging mutations in the methyl CpG-binding protein 2 (*MECP2*)[1]. Most evidence indicates that MECP2 regulates transcription globally after binding to methylated (m)CG dinucleotides in immature neurons and mCA dinucleotides in adult neurons[2–8]. Specifically, recent models support a role for MECP2 binding to mCA/mCGs in the gene-body that can: 1) slow RNA polymerase II elongation[5] 2) inhibit transcription initiation by looping interactions between the promoter and high-density MECP2-bound gene-bodies[9] or 3) inhibit transcription initiation by microsatellite interruptions of nucleosome binding that impact intragenic enhancer activation[10].

The number or fraction of mCA/mCG and gene-length have been primarily used to interpret transcription rate dysregulation in RTT mouse models[5,9]. However, the informative power of these DNA

features is limited, and it is still challenging to anticipate a priori which genes are transcriptionally up- or down-regulated in RTT, raising questions about whether other sequence features might also participate in transcription dysregulation mediated by the loss of MECP2[11]. Recent developments in machine learning techniques have revealed unsuspected DNA and RNA-sequence features associated with gene regulatory programs[12,13]. Employing these techniques in the RTT context could help explain the molecular mechanisms of MECP2 function and uncover other DNA sequence features important for MECP2 function.

Genome-wide analyses of steady-state mRNA levels and transcription rate changes in RTT models have demonstrated a global dysregulation of gene expression[5,9,14–18]. It is uniformly acknowledged that the magnitude of mRNA steady-state level changes is surprisingly

[1]Developmental & Stem Cell Biology, Hospital for Sick Children, Toronto, ON M5G 0A4, Canada. [2]Department of Molecular Genetics, University of Toronto, Toronto, ON M5S 1A8, Canada. [3]Genetics & Genome Biology, Hospital for Sick Children, Toronto, ON M5G 0A4, Canada. [4]Molecular Medicine, Hospital for Sick Children, Toronto, ON M5G 0A4, Canada. [5]RNA Bioscience Initiative and Department of Biochemistry & Molecular Genetics, University of Colorado School of Medicine, Aurora, CO 80045, USA. [6]These authors contributed equally: Deivid C. Rodrigues, Marat Mufteev. ✉e-mail: jellis@sickkids.ca

small[3,4,14,18,19]. In 2017, Johnson et al. [17] used GRO-Seq for nascent RNA and RNA-Seq of chromatin, nuclear, and cytoplasmic subcellular fractions to reveal the small steady-state alterations in the *Mecp2*-null mouse brain is the result of a previously unsuspected post-transcriptional regulatory mechanism. They proposed that large transcription rate changes are compensated by reciprocally adjusting mRNA half-life, and they provided initial support for the role of two RNA-binding proteins (RBPs). In particular, they examined the enrichment of 12 RBP binding sites in subsets of mRNAs and identified HuR-(ELAVL1) *cis*-acting elements in the 3′UTR with extended mRNA stability, or AGO2 *cis*-acting elements with reduced stability to implicate the action of unknown micro(mi)RNAs[17]. Their model of post-transcriptional regulation is similar to transcription buffering where RBPs present in the nucleus tag nascent mRNAs and shuttle with them to the cytoplasm. The RBPs then modify half-life to buffer transcription rate changes that preserve steady-state levels (reviewed by Hartenian E et al.[20]). These results have not been independently tested in mouse and it is unknown whether the mechanism is conserved in human RTT neurons. Therefore, it is important to experimentally measure the direction and magnitude of half-life changes in human *MECP2*-null neurons. Moreover, the post-transcriptional mechanism has not been studied by systematic enrichment analysis of all known miRNA and RBP *cis*-acting elements of the post-transcriptionally regulated mRNAs.

Here, we simultaneously investigate the potential role of sequence features mediating transcriptional dysregulation in RTT and expand on the post-transcriptional findings of Johnson et al. in human isogenic induced Pluripotent Stem Cell (iPSC)-derived RTT neurons. We use RNA-approach to equilibrium-sequencing (RATEseq) to measure transcription rate and half-life changes and employed machine learning to uncover sequence features underlying these changes in human and mouse RTT models. In parallel, we compare our human neuron findings to the high-confidence RNA-seq from subcellular fractionations of *Mecp2* mutant mouse brains[9]. We find that transcription rate changes in both human and mouse datasets are best predicted by combinations of three dinucleotide frequencies in gene-bodies that include the expected CA/CG motifs, but are most accurate if they also include other dinucleotides. We discover extensive half-life changes that identify: 1) a gene set with exclusive mRNA stability dysregulation (half-life only) and no associated transcription rate changes; and 2) a larger buffered gene set in which half-life regulation compensates for transcription rate changes that fully offset or minimize mRNA steady-state changes. We demonstrate a global absolute downregulation of miRNA levels, that corresponds with a global absolute half-life increase in RTT neurons. We find individual enriched miRNA binding-sites in the half-life only gene set but very few in the buffered gene set. RBP-binding sites were enriched in the 3′UTRs of half-life only genes, and distinct sites were also enriched in buffered genes with increased transcription rate. Overall, we propose that transcription rate increases in *MECP2* neurons are subject to surveillance by RBPs that post-transcriptionally regulate RNA half-life. We find that the buffering of transcription rate changes by half-life changes is a conserved feature of RTT models which minimize the steady-state changes in mRNA levels.

## Results

### Transcription rate changes in RTT neurons do not always alter mRNA steady states

To simultaneously measure direct changes in transcription rate and mRNA half-life in RTT neurons, we performed RATEseq on human cortical neurons derived from WT (NEU$_{WT}$) and *MECP2*-Null (NEU$_{RTT}$) isogenic patient-derived iPSCs[21] (Fig. 1a). Conventional RATEseq measurements of absolute half-life rely on the 4sU saturation curve. In addition, half-life fold-change can be derived by a second method that calculates a ratio between the steady-state value and the transcription rate of nascent RNA measured at early time points of 0.5 and 1 hour

(Fig. 1a). We use the term steady-state here in a restricted context to refer to the equilibrium between transcription rate and half-life in the whole cells at 24 hours.

In brief, iPSCs were differentiated into cortical neurons using a dual-SMAD protocol (Fig. S1A) and the non-neuronal cells were depleted based on the presence of specific surface proteins using MACS[22,23]. 4sU RNA labeling was performed and samples taken from 0.5 to 24 hours (except for the 1-hour NEU$_{RTT}$ samples). Cellular viability was consistently high as expected[24] (Fig. S1B, C). The steady-state was determined using 24-hour input samples that were treated with 4sU but without biotinylation and pulldown. Multiple RNA spike-ins (Fig. 1a) controlled for 4sU pulldown efficiency, background contamination, equivalent cell numbers, and sequencing quality[25], and indicate the high quality of the RNA samples and sequencing data (Fig. S1D–J). 3′end RNA-seq (QuantSeq) was utilized to quantitatively map 3′UTR isoform diversity to better understand the role of miRNAs and RBP-binding sites in half-life regulation and buffering of transcription dysregulation. Western blots of separate batches of the cells confirmed the absence of MECP2 in the NEU$_{RTT}$ samples (Fig. 1b). Efficient differentiation into mixed cortical neurons was confirmed by analyzing the steady-state mRNA abundance of a panel of neuronal marker genes (Fig. 1c) and comparing it to previously published transcriptomics of pluripotent stem cell-derived neurons of equivalent age or neurons collected from human fetal neocortex[26–28]. We first calculated the half-life using the 4sU saturation curve method[25] and deduced the transcription rate using the 0.5-hour time-point (plus a pseudoreplicate from the 1-hour time point for the NEU$_{WT}$, see methods) by calculating the number of newly synthesized mRNAs over 30 mins. Importantly, the measurement of transcription rate fold-changes between NEU$_{WT}$ and NEU$_{RTT}$ were highly comparable using only the 0.5-hour replicates or the 0.5 plus 1-hour time-point pseudoreplicates (Fig. S1K). However, the saturation curve-derived half-life measurements failed for most genes that had low transcription rates (Fig. S1L, M) and these are genes that may be repressed by MECP2 in neurons. We, therefore, calculated the relative half-life fold-changes using the ratio method, and these findings were similar to the saturation curve values for genes that it had determined with high confidence (Fig. S1N). Having shown the improved utility of the ratio method for the genes of most interest, we proceeded to next examine changes in transcription rate and how they correlate with changes in the steady-state.

As expected, we found widespread dysregulation of transcription rate and steady-state in the NEU$_{RTT}$ neurons (Figs. 1d–f, S1O, Supplementary Data 1). An independent 5-Ethynyl Uridine (EU) metabolic incorporation assay followed by qRT-PCR experimentally validated transcription rate changes of specific genes measured using both 0.5-hour or 0.5 and 1-hour time-points (Figs. 1g and S1P). Reassuringly, a comparison of our transcription rate datasets with a MECP2 ChIP-seq in mouse brain revealed that the genes with the highest changes in TR were more enriched for MECP2-binding (Fig. S1Q), and most transcriptionally upregulated genes displayed lower basal transcription rate in the WT controls (Fig. S1R)[9]. Importantly, we found that approximately half of the transcription rate dysregulated genes in NEU$_{RTT}$ were not altered at the steady-state mRNA level (Fig. 1h).

Given the relevance of these findings to disease mechanisms, we validated the discrepancy between transcription rate and steady-state changes in an orthogonal in vivo RTT system. We re-analyzed the high-confidence datasets from Boxer et al. [9] that sequenced nuclear and chromatin-associated mRNA abundance as a proxy for transcriptional dysregulation and the whole-cell fractions from cortical forebrain samples of WT, *Mecp2 y/-*, and point-mutant *Mecp2 R306C* adult mice (Fig. 1i). As shown previously in the context of in vitro-derived neurons, sequencing of nuclear RNA fractions is highly correlated to the transcription activity measured using GRO-seq[17]. Our analysis shows a similar pattern where approximately half of the genes transcriptionally dysregulated in the nucleus are not altered at the whole cell level that

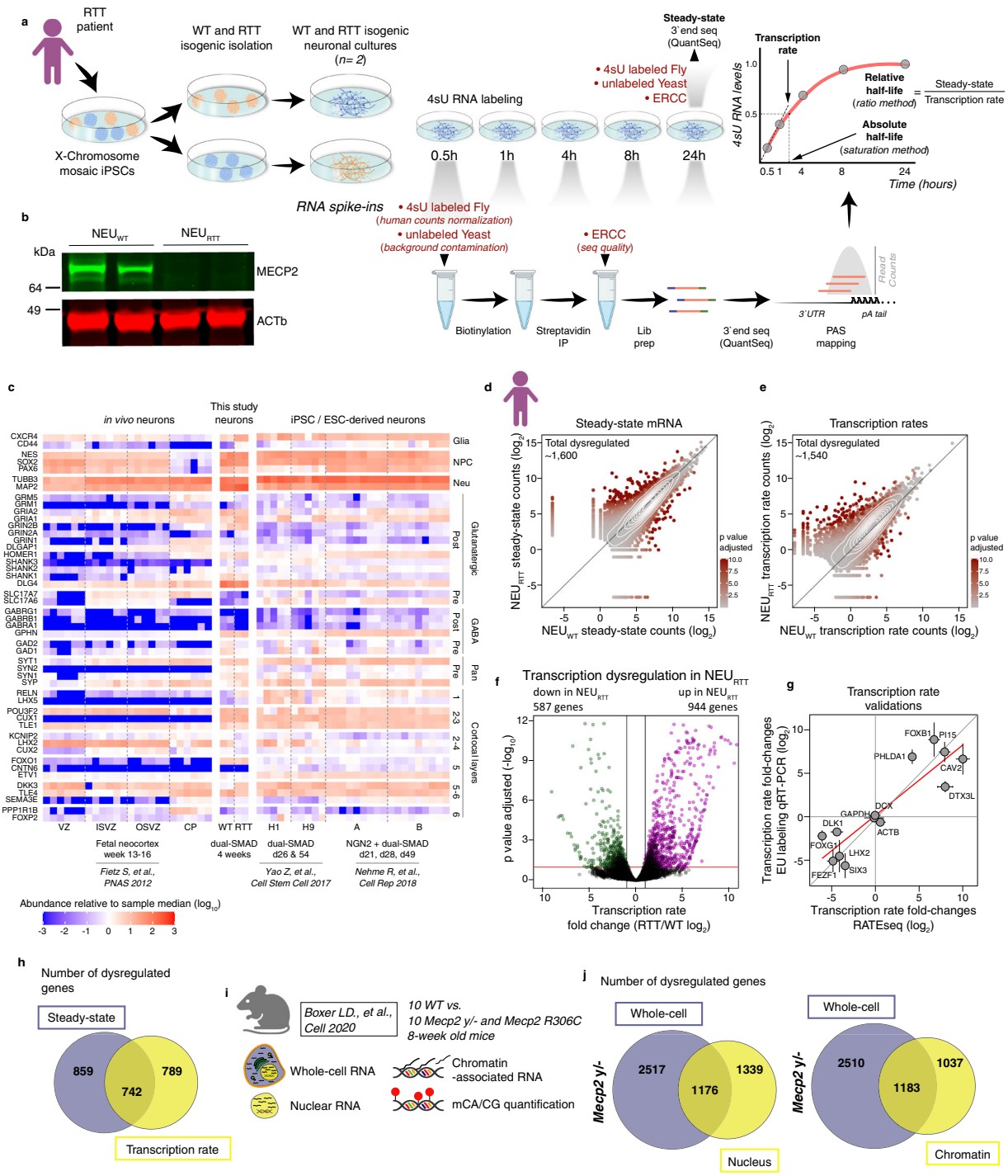

we define as the steady-state (Fig. 1i, j). This is consistent with the human neuron RATEseq findings that transcription rate dysregulation in the absence of MECP2 does not automatically result in altered steady-state mRNA levels, and that unaltered steady-state mRNA level does not automatically mean there is no change in transcription rate.

## The direction of transcription rate changes in RTT neurons is predicted by gene-body dinucleotide frequencies

The high fraction or number of mCA/mCG in longer genes have been associated with the small magnitude upregulation of these genes in *Mecp2* mouse models[3,7,9]. However, consistent with Johnson et al.[17], our analyses did not show a gene-body length effect in genes dysregulated at the transcription rate or steady-state (Fig. S2A). The contribution of other sequence features that predict which genes will be

transcriptionally dysregulated in RTT models has not been systematically evaluated in vivo[11]. To learn sequence features of the differentially expressed genes that are relevant for the direction of transcriptional shifts in our immature RTT neurons, we trained a classifier model using our measurements of genome-wide transcription rate changes in NEU_RTT (Figs. 2a and S2B). As anticipated by the lack of correlation between gene-body length and transcription rate changes (Fig. S2A), this model also resulted in random predictions based on gene-body length (including introns) (Figs. 2b and S2C, D). In contrast, frequencies of dinucleotides in the gene-body produced high prediction accuracies similarly found when using either the coding sequence (CDS) or 3′UTR (Fig. S2C, D). The prediction accuracies of CA/CG in gene-bodies were lower than predictions based on any combination of dinucleotides even when combined with gene-body length. This

**Fig. 1 | Changes in transcription rate in RTT neurons do not automatically result in altered mRNA steady states. a** Schematics of experimental outline for simultaneous quantification of transcription rate, mRNA half-life, and steady-state mRNA level. An isogenic pair of human WT and *MECP2*-null iPSC-derived cortical neurons were pulse-labeled with 4sU, and at designated time-points total RNA was harvested. 4sU-labeled *Drosophila melanogaster* (fly) and unlabeled *Saccharomyces cerevisiae* (yeast), and ERCC spike-in RNAs were added as indicated and used as pull-down efficiency, non-specific binding, library preparation, and sequencing controls. Steady-state mRNA levels were quantified from an aliquot of the 24 hour time point (non-biotinylated and unprocessed). This experiment was repeated for a total of two replicates. **b** Typical western blot showing the presence of MECP2 protein in two replicates of the NEU$_{WT}$ and absence in NEU$_{RTT}$ neurons. Uncropped western blot provided as a Source Data file. **c** Heatmap of mRNA abundance of cortical neuronal patterning and synaptic marker genes. The heatmap compares the mRNA abundance in the steady-state neuron samples generated in this study compared to neurons from previously published studies. The mRNA abundance in

each replicate is measured relative to the sample median. **d**, **e** Scatter-plots depicting genome-wide changes in steady-state and transcription rate. This experiment was repeated for a total of two independent replicates. **f** Volcano plot showing genes with increased or decreased transcription rate in *MECP2*-null neurons (NEU$_{RTT}$). **g** Transcription rate fold-changes determined by RATEseq (X-axis, standard error of the log2 fold-change estimated by DESeq2 from 4 biological replicates) were validated using an alternative approach. Neurons were incubated with 5-ethylnyl uridine (EU) and quantified following Click-it reaction and qRT-PCR (Y-axis, standard error of the log2 fold-change derived from qRT-PCR experiments using 3 biological replicates, *n* = 3) of genes selected to cover a large spectrum of fold-changes including genes with no changes. **h** overlap of genes altered at transcription rate and/or steady-state in human NEU$_{RTT}$. **i** Summary of samples from Boxer et al. re-analyzed in our study. **j** Overlap of genes with altered mRNA abundances in the nucleus or chromatin (transcription proxy) and whole-cell (steady-state) in the brains of *Mecp2* y/- mouse model. Panels **a**, **d**, and **i** were created with BioRender.com.

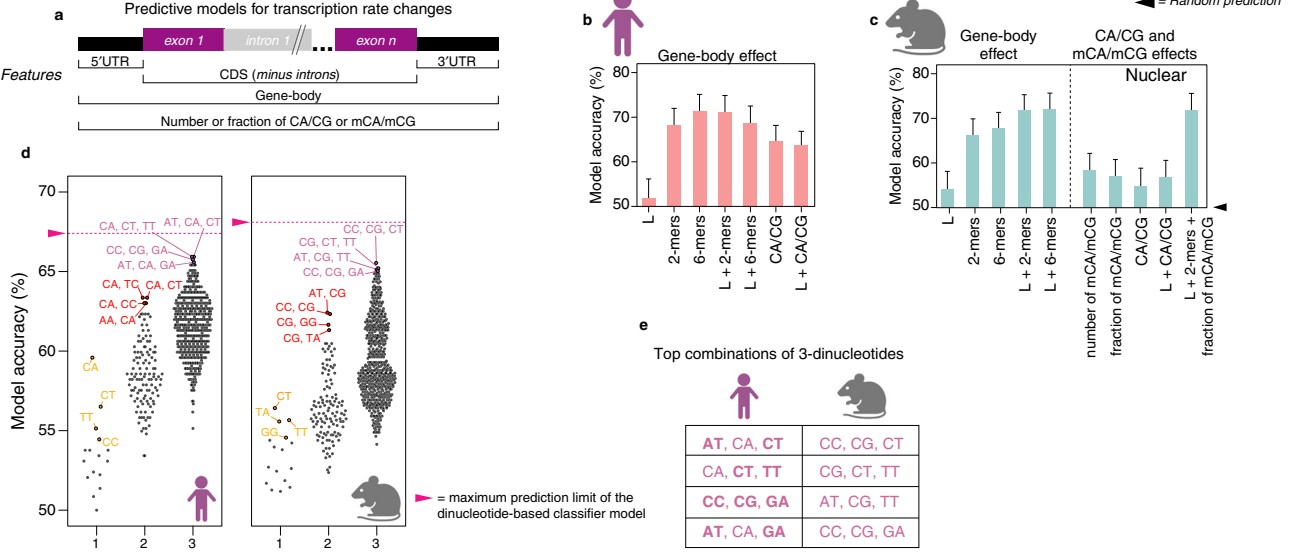

**Fig. 2 | The direction of transcription rate changes in RTT neurons is predicted by three combined gene-body dinucleotide frequencies. a** Random forest classifier for prediction of gene-body sequence features relevant for transcription rate fold-changes in human NEU$_{RTT}$ and in cortical brain samples of the mouse *Mecp2* y/-. Percent accuracy (Y-axis) of transcription rate fold-change predictions in human NEU$_{RTT}$ (**b**) or mouse *Mecp2* y/- (**c**) considering different gene-body sequence features. L = gene-body length; 2-mers and 6-mers= 2 and 6-nucleotide sequence elements, respectively; CA/CG = number of CA or CG di-nucleotides; mCA/mCG= methylated CA and CGs; + sign denotes combinations of two or more sequence features. Error bars depict 95% CI and are estimated from approximately 400 genes for human and 500 for mice. Log2 fold-change for each gene is derived from 4

independent replicates (30 minutes and 1 h) for human and 10 replicates for mice. **d** Top combinations of dinucleotides contributing to the full prediction accuracy described in **b** and **c**. Accuracy increases to the maximum when a specific combination of three dinucleotides is used. Similar behavior is observed between human (left) and mouse (right) datasets. Each point (accuracy) is estimated from approximately 400 genes for human and 500 for mice. The log2 fold-change for each gene is derived from 4 independent replicates (30 minutes and 1 h) for human and 10 replicates for mice. **e** Top combinations of three dinucleotides contributing to the full prediction model. Predictive di-nucleotides conserved between human and mouse are denoted in bold fonts in the human column. Panels **b**, **c**, **d**, and **e** were created with BioRender.com.

supports a prominent role for additional gene-body dinucleotide combinations in modulating transcription rates (Figs. 2b and S2C, D).

To test the combined dinucleotide-based predictive model in an orthogonal in vivo system, we trained a classifier model on the data from Boxer et al. which also includes cytosine methylation quantification (Fig. 1i)[9]. In the adult mouse brain, our model captured gene-body length and mCA fraction as predicting whether a gene is transcriptionally up-regulated in the absence of *Mecp2* (Fig. S2E, F). However, these models do not discern whether a gene is transcriptionally down-regulated, nor discriminate up- versus down-regulated genes (Fig. S2E, F). In contrast, the gene-body dinucleotide frequencies captured the direction of most transcriptional dysregulation in the *Mecp2* y/- mouse, with lower accuracy predicted by the CDS and UTR separately, an effect that was independent of gene-body length (Figs. 2c and

S2G–I). Surprisingly, combining CA/CG frequencies had less predictive accuracy than combinations of remaining dinucleotides (2-mers) independent of their methylation status (Figs. 2c, S2G–I). To investigate which dinucleotide or combinations thereof were responsible for the high predictive accuracy of transcription rate changes in RTT neurons, we repeated the classifier model considering single or multiple dinucleotide combinations. The classifier found that specific combinations of three dinucleotides reached the predictive accuracy of the full model for both human and mouse even though the specific combinations of three dinucleotides were different across species (Fig. 2d, e). Taken together, the classifier models of fetal stage human neurons and adult mouse brain indicate that combined frequencies of three dinucleotides that include non-CA/non-CG dinucleotides contribute to the direction of transcription rate modulation mediated by MECP2.

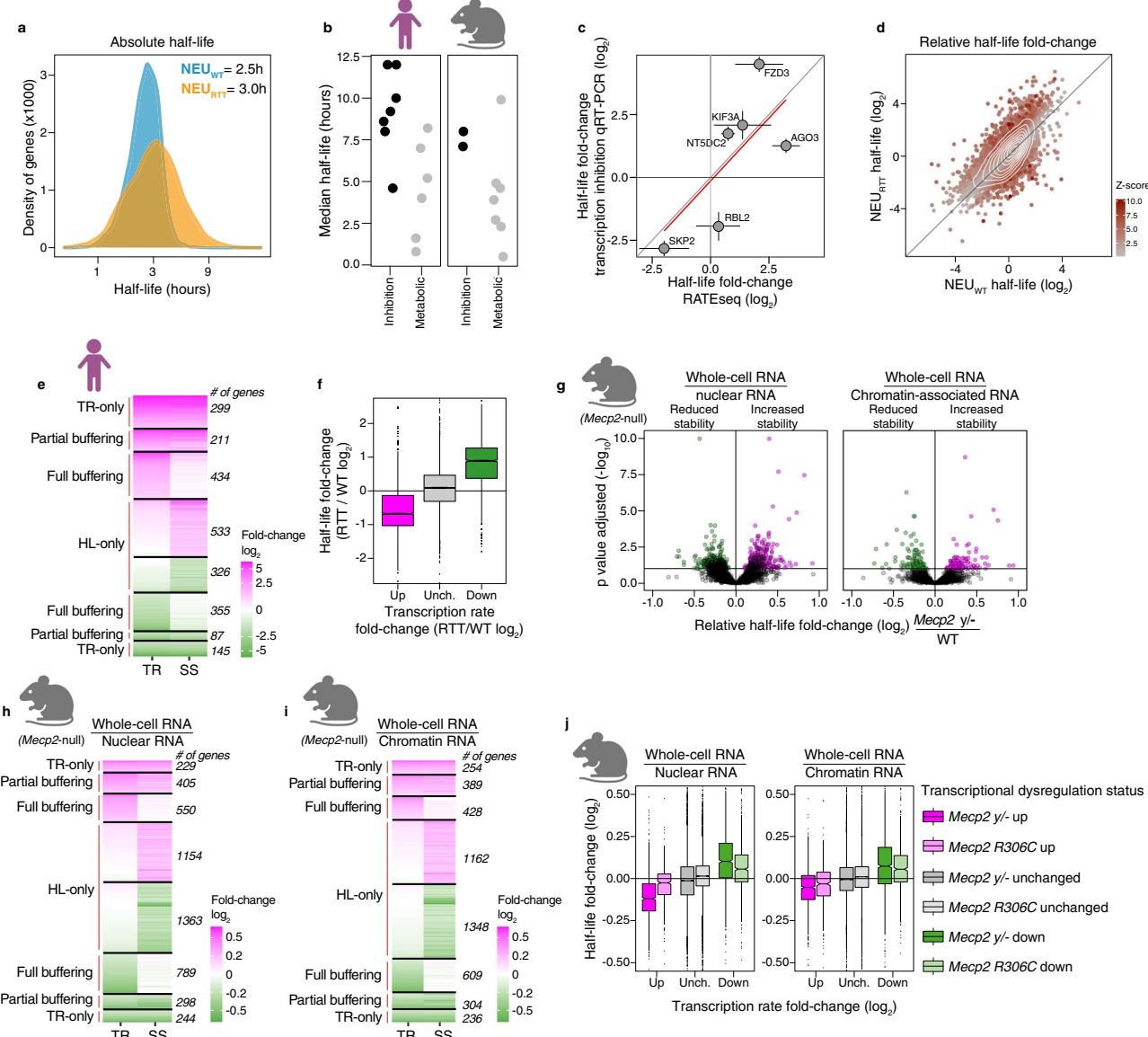

**Fig. 3 | Widespread changes in mRNA half-life directly alter steady-state abundance or buffer transcription rate changes in human and mouse RTT models. a** Global increase in the absolute half-life in NEU$_{RTT}$. Median half-life increases to 3 hours in NEU$_{RTT}$. **b** Median mRNA half-life, as measured in previous studies in human and mouse models, is higher than 1 hour (see also Supplementary Data 3). Black and grey dots: average half-life measured using transcription inhibition or metabolic incorporation methods, respectively. **c** Half-life changes determined by RATEseq (X-axis, saturation curve 2 biological replicates) were validated using transcription inhibition. Selected genes covering a wide range of fold-changes were quantified by qRT-PCR (Y-axis, 3 independent biological replicates). Error bars= spread of half-life log2 fold-changes measurement calculated from 50% CI of each genotype. **d** Global relative half-life changes in NEU$_{RTT}$. **e** Number of genes with changes in transcription rate only, partially or fully buffered by half-life changes, and genes whose change in steady-state are caused by altered half-life only (altered transcription rate= log2 fold-change p-adjusted value smaller than 0.1). Genes were defined with transcription rate only changes when the difference between steady-state and transcription rate log$_2$ fold-change was less

than 25% of transcription rate log$_2$ fold-change. **f** Most genes with increased (magenta) or decreased (green) transcription rate show decreased or increased half-life, respectively, and genes with unchanged transcription rate do not display significant changes in half-life. Transcriptional changes derived from data represented in Fig. 1f. **g** Global changes in relative half-life in brains of the *Mecp2* y/- mouse estimated from nuclear (left) or chromatin-associated RNAs (right) (horizontal line= FDR 0.1). **h, i** Number of genes with changes in nuclear or chromatin-associated RNA abundances (transcription rate proxies) partially or fully buffered by mRNA stability mechanisms, and genes whose change in steady-state are caused by altered half-life only. **j** *Mecp2* y/- and 306C mouse models display the reciprocal behaviour in half-life regulation relative to transcription rate. Fold changes derived from data represented in **h, i** (10 biological replicates). TR = transcription rate, HL = half-life, SS = steady-state. For **f** and **j**, up and low hinges are 25$^{th}$ and 75$^{th}$ percentiles. Up and low whiskers are 1.5*IQR (inter-quartile range) above and below the corresponding hinges. Notches are 1.58*IQR / sqrt(n) which matches 95% CI for median comparison. Panels **b**, **e**, **g**, **h**, **i**, and **j** were created with BioRender.com.

## Half-life fold-changes in RTT neurons directly alter the steady-state or buffer transcription rate

To measure changes in absolute half-life in hours we estimated the time required to reach half of the steady-state mRNA abundance from the 4sU saturation curves (saturation method, Fig. 1a). This absolute RATEseq half-life measurement revealed widespread changes in mRNA

stabilities that caused a global absolute increase in the median half-life from 2.5 to 3.0 hours in NEU$_{RTT}$ (and an increase in the mean half-life from 2.9 hours in NEU$_{WT}$ to 4.6 hours in NEU$_{RTT}$) (Figs. 3a, S1E–H, and Supplementary Data 2). The global half-life values for neurons correlate well with published median half-lives from other cell types (Fig. 3b, and Supplementary Data 3). These measurements were independently

validated on specific genes using a pulse treatment with actinomycin D to induce transcription inhibition followed by qRT-PCR (Fig. 3c). Other orthogonal 4sU pulse-chase methods for measuring half-life have high reproducibility compared to RATEseq[29]. Using the ratio method to measure relative half-life fold change, we found approximately 860 (0.5 h plus 1 h time-points) or 957 (0.5 h time-point only) genes (~35% or 30% of all genes with altered half-life, respectively) with significantly increased or decreased mRNA half-life but with unchanged transcription rate, termed half-life only (HL-only), leading directly to changes in the steady-state levels (Figs. 3d, e and S3A, B. See also Fig. 1h). The class of RNAs and their identities were highly comparable when we used incorporation rate data from either 0.5 h or 0.5 h plus 1 h to calculate transcription rates (Fig. S3B, C, and Supplementary Data 4) supporting the robustness of our calculation methods. These analyses show that a significant fraction of genes dysregulated at the steady-state level were exclusively driven by changes in mRNA stability.

We then explored the impact on steady-state mRNA levels when both half-life and transcription rate were changed. We found that in these cases, half-life moved in the opposite direction to the transcription rate and decreased the net change in steady-state levels (Fig. 3e–f). We found that 789 genes exhibited full buffering where the RNA half-life entirely offset the transcription rate change resulting in no steady-state change. A further 298 genes showed half-life changes that partially counteracted the transcription rate changes modulating steady-state changes. The contribution of half-life to the steady-state level changes (Fig. S3D) and buffering (Fig. S3E) was independent of the fold-change or FDR thresholds chosen. None of the measured changes correlate with gene-body or processed transcript lengths (Figs. S2A and S3F). Highly similar proportions of buffered and half-life only genes in NEU$_{RTT}$ were observed when calculations used the 0.5 h time point only on genes whose transcription rates were altered by at least 4-fold (Fig. S3B). Moreover, genes with high transcription rates were buffered by changes in half-life, showing that the buffering effect was not induced by detection noise from low abundance transcripts (Fig. S3G). The buffering effect was also observed when we used the saturation method (Fig. S3H, I). Altogether, these data indicate that the buffering of transcriptional dysregulation by opposite changes in mRNA half-life is a robust result not sensitive to the calculation method (ratio vs. saturation), signal-to-noise ratios, or the number of time points. We found ~28% of all genes ($n = 444$) with altered steady-state expression to be exclusively dysregulated at the transcription rate level (TR-only, Fig. 3e), underscoring the role of post-transcriptional regulation of mRNA stability in directly altering or buffering the RTT transcriptome.

**Proxy values support half-life shifts and transcription buffering in *Mecp2* mouse models**

Given the importance of finding global changes in half-life in the NEU$_{RTT}$ neurons and its implications for interpreting steady-state level dysregulation typically found in RTT models, we investigated whether these results were reproducible in vivo. To determine proxies of half-life changes in the mouse brain in RTT models we re-analyzed the data from Boxer et al. for whole-cell, nuclear, and chromatin-associated RNAs[9]. We accomplished this by comparing the abundance of transcribed genes in the nucleus and chromatin fractions as proxies for transcription rate as previously demonstrated[17], to the whole-cell fraction that includes mRNA undergoing decay in the cytoplasm[17] that we define as the steady-state (see methods). By applying the proxy value in the ratio method, we calculated relative half-life fold-changes. From the 5032 genes reported by Boxer et al. to be differentially regulated in *Mecp2* y/- mice, we found a similar pattern of widespread fold-changes in half-life (Fig. 3g, Supplementary Data 5). We also found similar proportions of HL-only genes (Fig. 3h, i), and the extent of full or partial buffering in both mouse models (Figs. 3j and S4A. See also Fig. 1h, j). In line with our RTT human neuron findings, the contribution

of half-life to the steady-state level changes (Fig. S4B) and buffering (Fig. S4C) was also independent of the fold-change or FDR thresholds chosen.

Despite the conserved relationship that half-life and transcription rate or their proxies have on steady-state gene expression in human and mouse RTT models, we found minimal overlap in the identities of genes dysregulated in each species (Fig. S4D, E). Furthermore, minimal overlap was also observed between the *Mecp2 y/-* and *Mecp2 R306C* mouse models as already observed by Boxer et al. (Fig. S4F, G). Overall, our results consistently identify steady-state level changes driven by half-life only without any measurable transcription rate shift in both human and mouse RTT models. Importantly, the majority of all half-life shifts fully or partially buffer *Mecp2*-mediated transcription rate dysregulation, and only a small number of genes have increased steady-state changes due to combined half-life and transcription rate changes in the same direction. Finally, similar to the human findings, we only find 473 genes (13% of total genes altered at steady-state, which excludes the full buffered group) with transcription rate only dysregulation as measured in the nuclear fraction. Altogether, our findings demonstrate a pattern of half-life shifts and transcription buffering that is conserved in RTT mouse and human models.

**Cis-acting elements in the 3′UTRs are highly associated with half-life changes**

Our findings indicate that RNA half-life is a critical regulatory layer defining steady-state RNA levels in RTT models. We therefore considered several potential mechanisms underlying how RNA half-life is controlled in RTT neurons: 1) alternative polyadenylation; 2) alternative-splicing; 3) codon usage integrating translation elongation to RNA stability; 4) sequence composition of 3′UTR and gene-bodies; and 5) enrichment of miRNA binding-sites and RBP *cis*-acting elements in the 3′UTR between buffered and non-buffered genes. Initially we investigated the contribution of mRNA isoforms by mapping and quantification of 3′UTR alternative poly-Adenylation events that showed no difference in the frequency of poly-Adenylation site usage in NEU$_{RTT}$ (Fig. S5A–C). Moreover, measurement of 3′UTR (NEU$_{WT}$ vs. NEU$_{RTT}$) and alternatively-spliced mRNA isoforms (mouse Boxer et al.) indicated that all mRNA isoforms display the half-life buffering effect to the same degree (Fig. S5D–F). These analyses argue against changes in mRNA isoform usage participating in the half-life shifts in RTT models.

Next, we created a classifier predictive model to estimate the effect of codons and sequence composition on the direction of changes to mRNA half-life (Fig. 4a). Overall 3′UTR length and nucleotide frequency had no predictive value for classifying increased or decreased half-life changes (Fig. 4b). Transcription rate changes were anti-correlated with half-life leading to high predictive accuracy. These results underscore the unidirectional and reciprocal link between transcription rate dysregulation and compensatory mRNA stability control in NEU$_{RTT}$. Dinucleotide frequencies in the 3′UTR offered significant prediction accuracy on whether the half-life was increased or decreased. Increasing the size of the tested k-mers from dinucleotides to 4-mers and 6-mers to encompass potential *cis*-acting elements improved the prediction accuracy of half-life (Figs. 4b and S5G), equivalent to transcription rate alone. A combination of transcription rate and 6-mers showed no further improvement in accuracy. Curiously, we found that classifier features in the CDS mirror that of the 3′UTR models also offering significant prediction accuracies for the half-life changes (Fig. S5H, I), highlighting a significant sequence composition correlation between CDSs and 3′UTRs (Fig. S5J). In contrast, comparison of the prediction models for in-frame codons (3-nt sequences) indicates that codon optimality has no effect on half-life changes in NEU$_{RTT}$ (Fig. S5I). Importantly, while the predictive accuracy of k-mers is lower in mouse, the classifier model predictions are upheld in the *Mecp2* mouse model. These findings show a reproducible

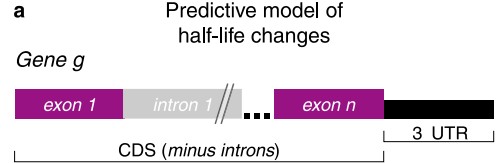

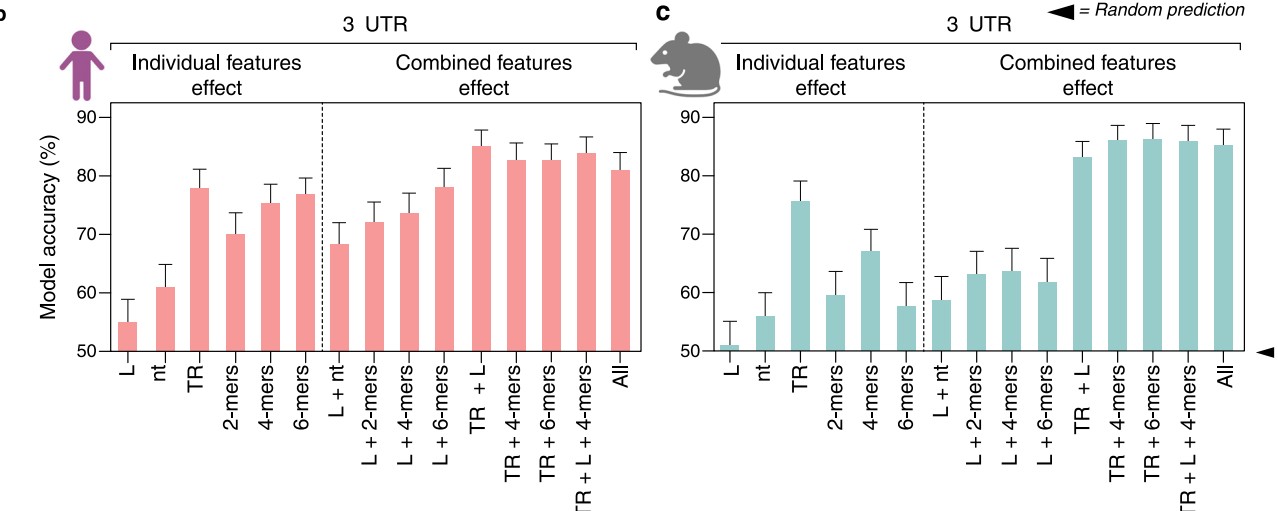

**Fig. 4 | *Cis*-acting elements in the 3′UTR are highly associated with half-life changes. a** Random forest classifier for prediction of mRNA sequence features relevant for half-life fold-change in human NEU$_{RTT}$ and in cortical brain samples of the mouse *Mecp2 y/-*. Percent accuracy (Y-axis) of half-life fold-change predictions in human NEU$_{RTT}$ (**b**) from 2 replicates or mouse *Mecp2 y/-* (**c**) from 10 replicates considering different mRNA sequence features. >80% prediction accuracies can be achieved with the features tested, and indicate that 3′UTRs contain sequence elements relevant for half-life changes in both humans and mice. L = gene-body length; nt= nucleotide sequence; 2-mers, 4-mers, and 6-mers= 2, 4, and 6-nucleotide sequence elements, respectively; TR = transcription rate; All= all features considered at the same time; + sign denotes combinations of two or more sequence features. Error bars depict 95% CI. Panels **b** and **c** were created with BioRender.com.

effect of transcription rate on half-life, thereby implicating a conserved buffering mechanism through *cis*-acting elements impacting half-life (Figs. 4c, S5K–M).

## miRNA and RBP *cis*-acting elements correlate with half-life changes in RNA stability exclusive genes

To examine a possible role of miRNAs in mRNA half-life regulation, we first performed small RNA-seq with a spike-in strategy to normalize cell numbers and inform on relative and absolute changes in miRNA abundance between the isogenic human NEU$_{WT}$ and NEU$_{RTT}$ (Fig. 5a). These results showed that the steady-state levels of miRNAs changed by as much as 4-fold up or down (Fig. 5b, Supplementary Data 6). Interestingly, most changes in miRNA steady-state abundance were captured in the RATEseq data by shifts in the transcription rate of miRNA genes (Fig. 5c, Supplementary Data 6). These findings indicate that transcription dysregulation of miRNA genes drives the changes in miRNA abundance, in addition to miRNA maturation processing as indicated previously[30,31]. Moreover, after normalization against a pool of small spike-in RNAs, the scaled absolute fold-change of most miR-NAs was decreased in NEU$_{RTT}$ (Fig. 5d–g). This global decrease in miRNA abundance in NEU$_{RTT}$ is in line with the global absolute upshift of median half-life (Fig. 3a).

To investigate the role of these miRNA changes in the regulation of half-life in NEU$_{RTT}$, we performed motif enrichment analysis of miRNA-binding sites present in the TargetScan database[32]. This analysis identified multiple potential miRNA-binding sites as enriched in up to 800 genes in the HL-only group of mRNAs whose steady-state level changes are exclusively directed by half-life changes (Fig. 5h, Supplementary Data 7). In contrast, we found significantly fewer

miRNA-binding sites sequences enriched in the group of buffered mRNAs with increased transcription rate and decreased half-life and many of these show no change in the miRNA abundance (Fig. 5i, Supplementary Data 7). We did not find miRNA sites enriched in the opposite group with decreased transcription rate and increased half-life (Supplementary Data 7). Overall, these data indicate that the reduced miRNA abundance contributes to the regulation of HL-only genes in NEU$_{RTT}$. However, very few individual miRNAs correlate with buffering, although combinatorial effects of multiple miRNAs cannot be excluded.

We then performed an unbiased search for the enrichment of 174 RBP *cis*-acting elements, as described in the RNACompete database[33], in the 3′UTR of mRNAs with altered half-life in NEU$_{RTT}$. We found hundreds of enriched RBP targets in the group of HL-only genes (Fig. 5j, Supplementary Data 7). This result includes the RBP HuR (ELAVL1) whose different motifs are enriched in 300-900 mRNAs with increased half-life in NEU$_{RTT}$. Our results suggest that 3′UTR-directed miRNA and RBP regulation best explain the HL-only gene set changes.

## Only RBP *cis*-acting elements are enriched in buffered genes with increased transcription rate and decreased half-life

Having excluded miRNAs as playing a substantial role in mRNA buffering, we explored a role for the 174 RBP *cis*-acting elements in the buffered group of mRNAs. We observed that no RBP *cis-acting* elements were enriched for genes with decreased transcription rate and increased half-life (Fig. 6a, right panel), and only a handful were depleted. This result suggests that RBPs are not involved in regulating transcripts with decreased transcription rate for half-life stabilization. In contrast, we found numerous RBP elements that were enriched in

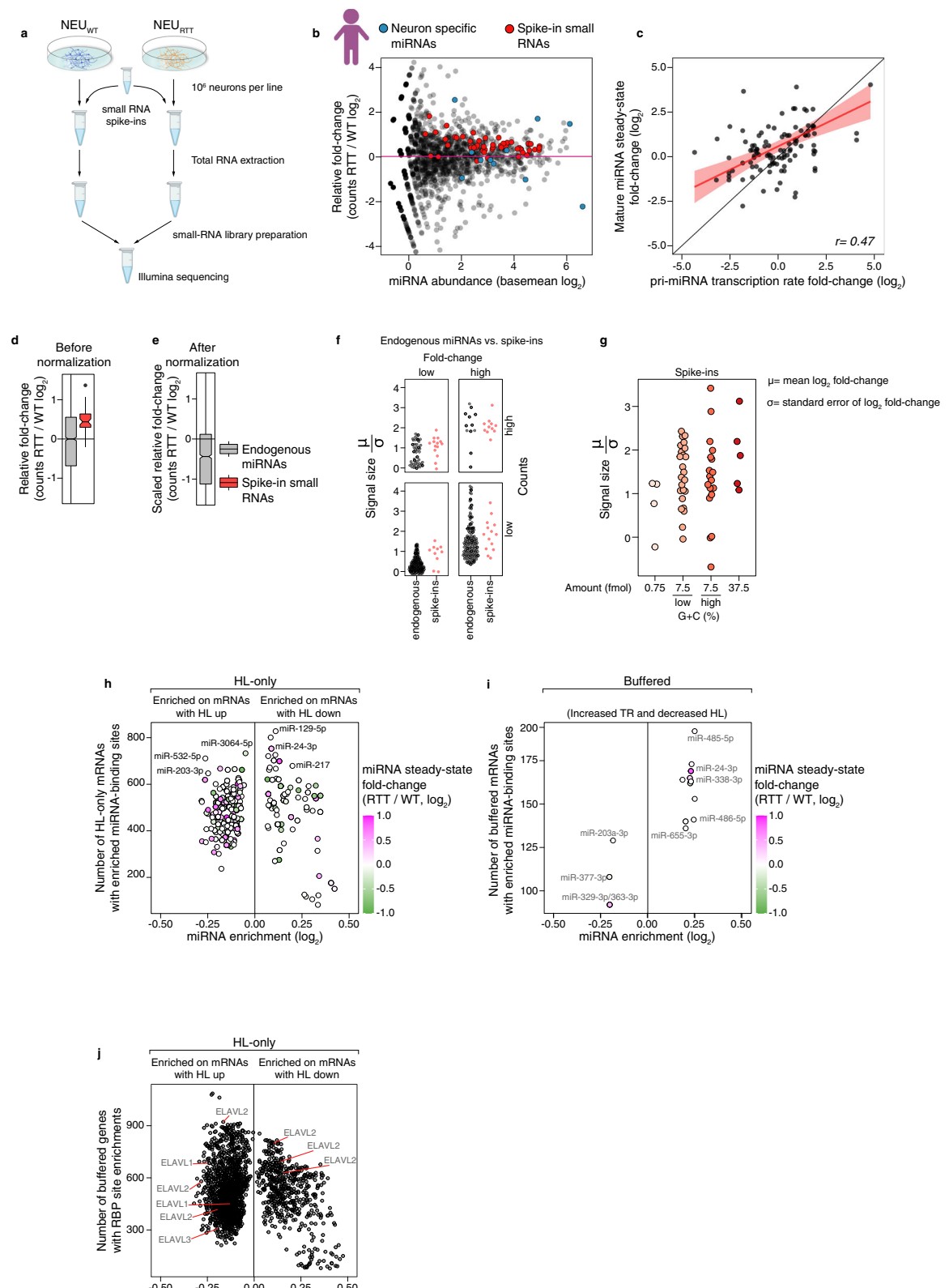

100-200 buffered genes with increased transcription rate and decreased half-life (Fig. 6a, left panel and Supplementary Data 7). To identify RBPs that plausibly regulate nascent mRNAs, we aggregated all the buffering enriched RBPs with reported cellular localizations, and found that 51 were nuclear and/or able to shuttle to the cytoplasm whereas only 18 were reported to be cytoplasmic only (Fig. 6b). To examine roles of specific RBPs, we first noted that ELAVL1 elements are

depleted in >150 genes in this set, demonstrating that it is only enriched in the HL-only gene set. In contrast, *NOVA2* and *ZFP36* were enriched in more than 200 buffered genes and these RBPs bind premature mRNA co-transcriptionally[34,35]. Additionally, *PTBP1* and multiple arginine-serine rich (SRSF) splicing factors that have been shown to participate in transcriptional buffering[36] were enriched in >150 genes. Further support for a directional role by *NOVA2* and SRSFs is that they

**Fig. 5 | miRNA and RBP *cis*-acting elements correlate with half-life changes in RNA stability exclusive genes. a** Small RNA-seq to quantify absolute changes in human miRNA abundances. Total RNA was extracted from the same number of NEU$_{WT}$ and NEU$_{RTT}$ neurons, to which we added the same mass of a small RNA spike-in mixture. This spike-in mixture contains small RNA molecules covering a wide range of sequence composition and molar concentrations. This experiment was repeated for a total of two independent replicates ($n = 2$). **b** Changes in miRNA abundances in the NEU$_{RTT}$. X-axis= basal abundance of each mature miRNA detected in the NEU$_{WT}$, Y-axis= fold-change in the NEU$_{RTT}$. Blue dots, neuronal-specific miRNAs accumulate at abundances higher than the mean in both NEU$_{WT}$ and NEU$_{RTT}$. Red dots, abundance of each small RNA spike-in added on a per-cell basis and used for library preparation control and absolute quantification of miRNA abundance. **c** Pearson's correlation of the transcription rate fold-changes between steady-state mature miRNA levels and primary miRNA (pri-miRNA) in NEU$_{WT}$ and NEU$_{RTT}$ showing a significant correlation between both ($r = 0.47$, p val $4.3^{-7}$), indicating many changes in mature miRNA steady-state levels are caused by changes in

their transcription rate. Error bars= 95% CI of the linear model. **d** DESeq2-calculated miRNA level fold-change before and after spike-in normalization (**e**). The absolute abundance of miRNAs in NEU$_{RTT}$ is reduced for most miRNAs. Up and low hinges= 25th and 75th percentiles. Up and low whiskers= 1.5 *IQR (inter-quartile range) above and below the corresponding hinges. Notches= 1.58 *IQR/sqrt(n) matching 95% CI for median comparison. **f–g** The precision of fold-change measurement is quantified as a ratio of $\log_2$(fold-change) and its standard error. Comparison with endogenous miRNAs and number of detected counts (**f**) or comparison between small RNA spike-in across molar concentrations and G + C content (**g**). miRNA-binding sites enriched in half-life only (**h**) or in combination with transcription rate up and half-life down (**i**). Y-axis= the number of genes containing miRNA-binding sites found, and X-axis= enrichment levels. Color represents the miRNA steady-state fold-change in the NEU$_{RTT}$ measured in **b**. **j** 7-mers known to be targeted by RBPs enriched in the HL-only group based on the RNAcompete database[33]. TR = transcription rate, HL = half-life. Panels **a** and **b** were created with BioRender.com.

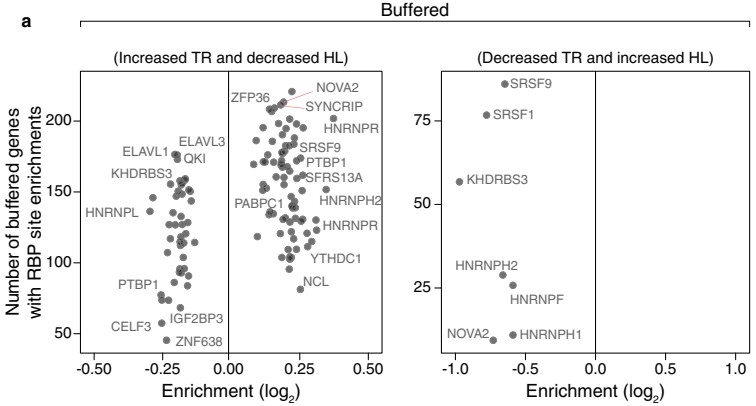

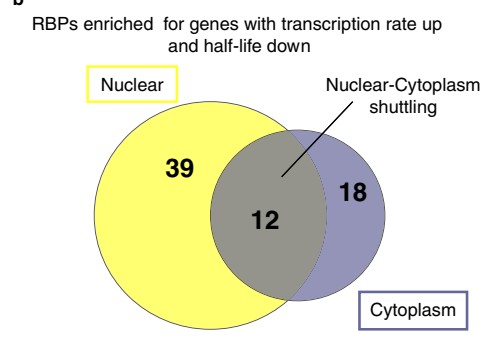

**Fig. 6 | RBP *cis*-acting elements are enriched in buffered genes with increased TR and decreased half-life. a** 7-mers known to be targeted by RBPs enriched in the group of buffered mRNAs. A different set of RBP *cis*-acting elements was found

enriched compared to the HL-only group (Fig. 5j). **b** Cellular distribution of the RBPs enriched in mRNAs with increased transcription rate and decreased half-life showing that these are predominantly nuclear.

were depleted in the buffered genes with decreased transcription rate and increased half-life (Fig. 6a). Our results reveal specific sets of RBP motifs associated with half-life regulation of buffered genes with increased transcription rate in RTT neurons.

## Discussion

Our analysis of a dataset of transcription rate and half-life changes in human neurons and independent re-analyses of mouse RTT models demonstrate that most transcription rate changes in the absence of *MECP2* are buffered by post-transcriptional regulation of mRNA stability. We used RATEseq to simultaneously measure transcription rate and half-life changes in human neuron samples. We complemented this approach by applying the subcellular fractionation strategy employed by Johnson et al. to the mouse resource dataset of Boxer et al. The consistent findings from both methods and model systems unambiguously show that post-transcriptional regulation is a modifier of transcription rate dysregulation in RTT, and that it is a conserved mechanism shared by human and mouse neurons. We provide evidence for changes in steady-state levels driven solely by half-life only shifts, and large transcription rate shifts that are entirely offset at the steady-state level by half-life mechanisms that we refer to as buffering. These observations have major implications for interpreting RNA-Seq results in RTT and potentially other neurodevelopmental disorders, or in diseases of other tissues caused by mutations in genes that modulate transcription like *MECP2*. The existence of transcriptional buffering mechanisms in mammals raises a cautionary note for interpreting RNA-Seq steady-state results in the general context of transcriptional regulation. It also argues in favour of more widely

prioritizing methods that directly measure nascent transcription or account for mRNA stability. Moreover, we extend the limited search by Johnson et al. who found two RBP *cis*-acting elements enriched in mRNAs with altered half-life by discovering hundreds of RBP *cis*-acting elements in the half-life only gene set. We also demonstrate the enrichment of miRNA-binding sites in the HL-only gene set. The buffered gene set with increased transcription rate likely describes the genes transcriptionally repressed by MECP2[37]. In this group, we identified a restricted subset of mostly nuclear or shuttling-capable RBPs whose *cis*-acting elements are highly enriched. Our findings thus reveal numerous candidate RBPs potentially involved in buffering the increased transcription rate of their mRNA targets. This mechanism may act as a network to coordinate mRNA degradation in healthy neurons and to compensate for transcription rate dysregulation in RTT neurons.

It is important to note that, while we found transcriptional buffering conserved between the human and mouse RTT models, the identity of genes dysregulated at both transcription and mRNA stability between species was considerably different. It is conceivable that the differences in gene identities between ours (human) and the Boxer et al. (mouse) datasets originate from the analysis of different cell types (in vitro-derived human neurons vs. mouse whole brains), or from the maturation stage differences between the studies. A high degree of variability between the dysregulated genes in the RTT mouse models has also been observed previously[38]. While the nature of this variability is, at this point, obscure, comparative studies have shown agreement on a core set of dysregulated genes[38]. Nonetheless, this indicates a high level of flexibility of the buffering mechanism.

Our computational methods were focused on defining *cis*-acting elements that are relevant for transcription rate or half-life regulation. At the DNA level, our classifier model discovered that the direction of transcription rate shifts in human and mouse RTT models are best predicted by combinations of three dinucleotides that include one or both of the canonical MECP2-binding sites CA and CG, together with other dinucleotides such as AT. The MECP2 AT-hook domain contributes to low-affinity transient interactions with AT-rich DNA that influence the dynamics of MECP2 binding to local high affinity methylated DNA sites[39]. Tellingly, the classifier predictions are low when modelling only CA/CG dinucleotide frequencies without accounting for low-affinity sites. These unbiased findings from machine learning algorithms indicate that the gene-body frequencies of other dinucleotides like AT are important and are a conserved mechanism defining the direction of transcription rate changes in RTT neuronal models. We speculate they may act by transient recruitment of MECP2 influencing its local binding dynamics to nearby methylated DNA. We confirmed a role of gene-body length in transcription rate regulation but only in the adult mouse neuron resource of Boxer et al. However, neither our classifier model on the human neuron RATEseq dataset nor the adult mouse dataset from Johnson et al. support this observation. It is possible that gene-body length contributes less to transcription rate regulation in fetal stage neurons derived from iPSC in which only mCG modifications are expected to be present[7,40], or that it requires the power of ten replicate samples used in the Boxer et al. resource to be detected.

With regards to half-life only regulation, to define which miRNA-binding sites to investigate we first used our human neuron miRNA dataset to identify the miRNAs that are altered in NEU$_{RTT}$. These results confirm the reported miRNA changes in RTT mouse models, although the RATEseq dataset shows that intergenic pri-miRNA transcription rate is altered in RTT adding another dimension to the known miRNA processing alterations in mouse[30] and human neurons[31]. Through the use of spike-in scaling in the miRNA dataset, we deduced a global absolute downregulation of miRNAs in NEU$_{RTT}$ that account for the global absolute increase in mRNA half-life of ~0.5 hours. Superimposed on the increased global half-life effect were individual HL-only genes which our classifier models revealed strong enrichment of miRNA-binding sites consistent with their known role in mRNA instability. Many RBP *cis*-acting elements including ELAVL1 (HuR) were enriched in the HL-only gene set. While Johnson et al. reported HuR and AGO2 *cis*-acting element enrichments in buffered gene sets in the mouse using the subcellular fractionation approach, our RATEseq results and unbiased search of RBP *cis*-acting elements point to a role for miRNAs and ELAVL1 in mRNA stability in human neurons rather than in the buffering mechanism itself.

While we eliminated several possible buffering mechanisms, one limitation is that we were unable to test a potential role of mRNA methylation modifications or poly(A)-tail length regulation on the targeted mRNAs with altered transcription rate[41]. We speculate that these mechanisms could also participate in the buffering mechanisms, particularly in the gene set with decreased transcription rate and increased half-life that we found was not associated with enrichment of either miRNA-binding sites or RBP *cis*-acting elements. The simplest mechanism for buffered genes with increased transcription rate is that the RBPs bind nascent mRNA in the nucleus and are transported to the cytoplasm where they tag the transcript for stabilization. We speculate that the nuclear RBPs are limiting in neurons, and if transcription rate increases then the proportion of tagged mRNA falls, leading to relatively more degradation in the cytoplasm and decreased half-life. A more complex variation on this model is that the concentration of some RBPs themselves may also change in RTT, and this may increase or decrease their ability to stabilize their target mRNAs. To distinguish these models, it will be necessary to determine which RBPs are changed at the protein level in RTT using proteomics of neuronal nuclei and

then individually testing their impact through gain- or loss-of-function assays on the mRNA targets.

Equivalent loss-of-function experiments have already been described in humans with neurodevelopmental disorders caused by mutations in RBP genes such as *NOVA2*[35]. The impact of these RBPs on buffering could be established using existing or new iPSC or mouse models. In fact, global transcription rate and RBP concentrations will inevitably be altered during the course of neurodevelopment, suggesting that it would be valuable to define the buffered gene sets in iPSC and their progeny Neural Progenitor Cells (NPC) relative to the final neurons described here. We and others[18,23] have previously reported translational regulation changes in RTT neurons in both ribosomal loading and protein stability implemented through alterations of E3-ubiquitin ligase protein levels. These findings emphasize that buffering in RTT and potentially other disorders likely operates at both the mRNA and protein levels.

## Methods

### iPSC cultures and neuronal differentiation

iPSC lines #37 (WT) and #20 (isogenic *MECP2*-null) from a female patient were previously described[21]. Both cell lines were generated and cultured under the approval of the SickKids Research Ethics Board and the Canadian Institutes of Health Research Stem Cell Oversight Committee. iPSC lines were cultured in 5% $CO_2$ on BD hESC-qualified matrigel (BD) in mTeSR medium (STEMCELL Technologies). Cultures were passaged using ReLeSR (STEMCELL Technologies) following the manufacture's instruction every 6-7 days. For neuronal induction, iPSCs were aggregated as Embryoid Bodies (EBs) in low-attachment dishes in N2 media containing laminin (1 ml/ml) with 10 mM SB431542, 2 mM DSM, and 1x penicillin-streptomycin changed daily. After 7 days, EBs were plated on poly-L-ornithine + laminin-coated dishes and grown in N2 media + laminin (1 ml/ml). After 7 days, neural rosettes were manually picked and transferred to poly-L-ornithine + laminin-coated wells. After 7 days, neural rosettes were picked a second time, digested with Accutase and plated on poly-L-ornithine + laminin-coated wells. Resulting NPCs were grown as a monolayer and split every 5-7 days in NPC media containing DMEM/F12, N2, B27, 1x non-essential amino acid (NEAA), 2 mg/ml Heparin, 1 mg/ml laminin. To generate neurons, NPCs were plated on poly-L-ornithine + laminin-coated plates at a density of $10^6$ cells per 10 cm dish and cultured for 3 weeks in neural differentiation medium (Neurobasal, N2, B27, 1 mg/ml laminin, 1x penicillin-streptomycin, 10 ng/ml BDNF, 10 ng/ml GDNF, 200 mM ascorbic acid, and 10 mM cAMP).

### Neuronal enrichment using MACS

Neuronal cultures were enriched for all experiments to exclude contaminating glia and NPCs present after differentiation. Enrichment of 3-week old neuronal cultures was made as described earlier[23,42]. 3-week old heterogeneous neuronal cultures were enriched by a negative selection strategy using antibodies against surface markers CD44 (Biotin Mouse Anti-Human CD44, 1:1000 dilution, BD Biosciences, Clone G44-26; Cat# 555477) and CD184 (Biotin Mouse Anti-Human CD184, 1:1000 dilution, BD Biosciences, Clone 12G5; Cat# 555973) recognizing NPCs, glial progenitors and astrocytes[43] using magnetic-activated cell sorting (MACS® - Miltenyi Biotec). After enrichment, neurons were re-seeded onto Matrigel-coated 6-well plates, cultured in neural differentiation medium, and allowed to recover for one extra week, for a total of 4 weeks neuronal differentiation.

### Western blots

Cells were washed in ice-cold PBS and total protein extracted in radioimmune precipitation assay (RIPA) buffer (25 mM Tris-HCl, pH7.6, 150 mM NaCl, 1% Nonidet P-40, 1% sodium deoxycholate, and 0.1% SDS). Equivalent protein mass was loaded on SDS-PAGE and transferred to Hybond ECL (GE HealthCare) nitrocellulose membrane.

Antibodies MECP2 (Rabbit Anti-MECP2, 1:1000 dilution, Millipore, Cat# 07-013; RRID:AB_2144004), and Beta Actin (Mouse Anti-bActin, 1:5000 dilution, Sigma-Aldrich, Cat# A5441; RRID:AB_476744) were used. Near-Infra Red-conjugated secondary antibodies (IRDye 800CW Donkey anti-Mouse IgG, Cat# 926-32212, and IRDye 680RD Donkey anti-Rabbit IgG, Cat# 926-68073, 1:25000 dilution for both, LI-COR) were used and membranes scanned using LI-COR Odyssey CLx scanner according to manufacturer's instructions. Acquired images were analyzed using ImageStudio v5.2.5.

### 4sU metabolic labeling of Neurons and RNA extractions

When enriched neuronal cultures reached 4-week of differentiation, media was replaced with neuronal differentiation media supplemented with 100 µM 4sU (Sigma-Aldrich) reconstituted in DMSO. Neurons were harvested at 0.5, 1, 4, 8, and 24 h after the addition of 4sU (except for the *MECP2*-null line where the time-point 1 h was omitted from both replicates due to low differentiation yields). Metabolic labeling was designed such that all time points were collected together. After incorporation, cells were quickly washed twice with ice-cold 1× PBS Total RNA and scraped into ice-cold 1.5 ml Eppendorf tubes. Cells were collected by spinning at 1000 $g$ for 5 min at 4 °C and cell pellets were resuspended in 1 mL of Trizol (Thermo Fisher Scientific). Total RNA was extracted according to manufacturer instructions. The steady-state sample was prepared from a 5 µg aliquot of the 24 h time-point added with 0.5 µg of both 4sU labeled and unlabeled spike-in RNAs. Neuronal viability in the presence of 100 µM 4sU was monitored up to 24 h of treatment on parallel cultures by using Trypan blue staining and live/dead cell counting.

### Biotinylation and pull down of 4sU-labeled RNAs

50 µg of total neuronal RNA was mixed with 5 µg unlabeled *yeast* RNA and 5 µg 4sU-labeled S2 *fly* RNA in a total volume of 120 µL. 1 mg/mL HPDP-biotin (ThermoFisher Scientific) was reconstituted in dimethylformamide by shaking at 37 °C for 30 min at 300 RPM. 120 µL of 2.5× citrate buffer (25 mM citrate, pH 4.5, 2.5 mL EDTA) and 60 µL of 1 mg/mL HPDP-biotin were added to the premixed RNA sample for each time point. The solution was incubated at 37 °C for 2 h at 300 RPM on an Eppendorf ThermoMixer F1.5 in the dark. Samples were extracted twice with acid phenol, pH 4.5, and once with chloroform. RNA was precipitated with 18 µL 5 M NaCl, 750 µL 100% ethanol, and 2 µL GlycoBlue (Invitrogen) overnight at −20 °C. Precipitated RNA was pelleted for 30 min at 21,000 $g$ at 4 °C. The RNA pellet was resuspended in 200 µL of 1× wash buffer (10 mM Tris-HCl, pH 7.4, 50 mM NaCl, 1 mM EDTA). Biotinylated RNA was purified using the µMACS Streptavidin microbeads system (Miltenyi Biotec). 50 µL Miltenyi beads per sample were pre-blocked with 48 µL 1× wash buffer and 2 µL yeast tRNA (Invitrogen), rotating for 20 min at room temperature. µMACS microcolumns were washed 1× with 100 µL nucleic acid equilibration buffer (Miltenyi Biotec), followed by 5× washes with 100 µL 1× wash buffer. Beads were applied to microcolumns in 100 µL aliquots and again washed 5× with 100 µL 1× wash buffer. Beads were demagnetized and eluted off the column with 2×100 µL 1× wash buffer, and columns were placed back on the magnetic stand. A total of 200 µL beads was mixed with each sample of biotinylated RNA and rotated at room temperature for 20 min. Samples were applied to the microcolumns in 100 µL aliquots, washed 3× with 400 µL wash A buffer (10 mM Tris-HCl, pH 7.4, 6 M urea, 10 mM EDTA) prewarmed to 65 °C, and washed 3× with 400 µL wash B buffer (10 mM Tris-HCl, pH 7.4, 1 M NaCl, 10 mM EDTA). RNA was eluted with 5× 100 µL of 1× wash buffer supplemented with 0.1 M DTT, and flow-through was collected in a tube. Purified RNA was precipitated with 30 µL 5 M NaCl, 2 µL GlycoBlue, and 1 mL 100% ethanol, incubated at −20 °C overnight. Samples were spun at 21,000 $g$ at 4 °C for 30 min and resuspended in 20 µL water. RNA quality was assessed by running 3 µL of samples on a 1.5% agarose gel.

### Transcription rate measurement using EU

Transcription rate measurements were validated by an alternative method using the metabolic incorporation of 5-ethynyl uridine (5-EU) followed by quantifying mRNA levels by qRT-PCR. NEU$_{WT}$ and NEU$_{RTT}$ were incubated with 0.5 mM 5-EU (ThermoFisher) for 30 min. Total RNA was extracted and processed using Click-iT Nascent RNA Capture Kit (ThermoFisher) according to the manufacturer's instructions. The captured RNAs were used as a template for cDNA synthesis, followed by qRT-PCR to quantify mRNA level (for the primer list, please see supplementary information file). Genes were chosen to cover a wide range of transcription rate changes determined by RATE-seq.

### Half-life measurements using transcription inhibition

Half-life measurements were validated by an alternative method using transcription inhibition followed by quantifying mRNA levels by qRT-PCR. 10 µg/mL actinomycin D (Sigma-Aldrich) was added to NEU$_{WT}$ and NEU$_{RTT}$. RNAs were isolated at 1 h, 3 h, and 6 h time points using the RNeasy Plus kit (QIAGEN). The RNAs were used as a template for cDNA synthesis followed by qRT-PCR to quantify mRNA levels (for the primer list, please see supplementary information file). Genes were chosen to cover a wide range of half-life changes as determined by RATE-seq. Fold-changes for all time points were calculated relative to the 0-hour time point. Then, data was fit with *lm* function from R (formula = log(Fold-change, 2) ~ Time + 0). The coefficient of "Time" term measures degradation rate $\beta$. And a half-life is derived as $\ln(2)/\beta$. Finally, the confidence intervals are estimated with the *confint* function (*stats* package).

### cDNA synthesis and qRT-PCR

cDNAs were synthesized using SuperScript III reverse transcriptase (ThermoFisher) with random hexamer primers according to the manufacturer's instructions. For qRT-PCR, we used SYBR Select PCR Master Mix (ThermoFisher). Fold-changes were calculated by the ΔΔCt methods using Glyceraldehyde 3-phosphate dehydrogenase (GAPDH) and 18 S as housekeeping genes, averaged between technical and subsequently biological replicates to achieve an average fold difference (for the primer list, please see supplementary information file).

### miRNA extraction and spike-in strategy

To calculate relative and absolute differences in the miRNA population in NEU$_{WT}$ and NEU$_{RTT}$, small RNAs were extracted from two replicates of both lines using the same number of cells followed by the addition of a set of spike-in RNAs. Small RNAs were extracted from 500,000 neurons of each line using the SPLIT RNA extraction Kit (Lexogen) according to the manufacturer's instructions. A set of 52 RNA spike-ins (QIAseq miRNA Library QC Spike-Ins – Qiagen) that spanned a wide range of concentrations were added to the recovered RNAs according to the manufacturer's instructions. Sequencing libraries were made using the Small RNA library preparation kit NEBNext (NEB) according to the manufacturer's instructions. Sequencing was performed on the Illumina HiSeq 2500 using the Rapid Run mode. Datasets can be accessed from GEO using the access number GSE191168.

### Library preparation and RNA-sequencing

RNA-seq libraries were prepared for each time-point and steady-state sample using the QuantSeq 3' mRNA-Seq Library Prep Kit FWD for Illumina (Lexogen) automated on the NGS WorkStation (Agilent) at The Centre for Applied Genomics (TCAG) according to the manufacturer's instructions. PCR cycle numbers were determined using the PCR Add-on Kit for Illumina (Lexogen). All steady-state samples were processed with 250 ng of total RNA input. To minimize variability between time-points within a batch, RNA samples were processed with the same total RNA input with a minimum of 100 ng of total RNA used. Each sample was spiked-in with ERCC RNA Spike-In Control Mix 1 (Ambion) according to the manufacturer's instructions prior to the

start of library preparation. Library quality and quantity were measured at TCAG prior to sequencing with Bioanalyzer (Agilent) and KAPA qPCR (Roche). Sequencing was also performed at TCAG on the Illumina HiSeq 2500 with single-end 100 bp read length yielding 40 to 50 million reads. Datasets can be accessed from GEO using the access number GSE191168.

### Processing of raw sequencing reads

Processing starts with trimming of reads in 4 steps using cutadapt version 1.10[44]. First, we removed adapters exactly at the 3′-end of the reads (-a AGATCGGAAGAGCACACGTCTGAACTCCAGTCAX -O 4 -e 0.1 --minimum-length 25). Second, we removed internal or long stretches of adapter (-a AGATCGGAAGAGCACACGTCTGAACTCCAGTCA -O 30 -e 0.18 --minimum-length 25). Third, we trimmed low-quality bases at the 3′-end of the reads (-q 20 -O 4). Finally, we removed poly-A tail at the 3′-end of the reads (-a AAAAAAAAAAAAAAAAAAAAAAAAAAAAAAAAX -O 4 -e 0 −minimum-length 25).

### Generation of custom hybrid genome index and reads alignment with STAR

We generated a custom genome index to accommodate the quantification of yeast, fly, and ERCC spike-in RNA. Annotations (gencode version 29, flybase version all-r6.22, saccharomyces_cerevisiae.gff from yeastgenome.org, custom for ERCC) and genomes (hg38, dm6, sacCer3, ERCC from ThermoFisher) for all species and ERCC were combined and then processed with STAR version 2.6.0c (--sjdbOverhang 100). Finally, reads are aligned to hybrid genome with STAR version 2.6.0c (default settings)[45].

### Quantification of RNA abundance

Poly-A sites were obtained from PolyA_DB version 3 and converted to hg38 coordinates with *liftOver* (UCSC)[46,47]. Reads with MAPQ < 2 are filtered out. Finally, usage of poly-A sites was defined as a sum of reads whose 3′-ends are falling within 20 bp upstream and 10 bp downstream of the poly-A sites. The sum was counted with a custom Python script using *pybedtools, pysam, pypiper*[48–50]. Annotation of pri-miRNA transcripts structures was downloaded from Mendel lab[51]. Each transcript was matched to miRNA gene based on overlaps with GENCODE annotated pre-miRNA coordinates[52]. Then, pri-miRNA poly-A sites overlapping mRNA or lncRNA poly-A sites from PolyA_DB were removed. Finally, usage of pri-miRNA poly-A sites was quantified with *featureCounts* (strandSpecific=1, read2pos = 3 from *Rsubread* package)[53]. The abundance of mature miRNAs was quantified with *mirdeep2* pipeline[54]. Reads were preprocessed and collapsed with *mapper.pl* script (-e -h -j -k AGATCGGAAGAGCACA -l 18 -m -v) and quantified with *quantifier.pl* script, using hairpin and mature sequences obtained from miRbase[55].

### Heatmap comparison of neurons

Counts of each replicate are normalized relative to the median of the sample and log-transformed. Transformed values are visualized for the NPC and Glia markers in addition to a list of neuron-specific genes obtained from Zaslavsky et al. [56].

### Normalization of human read counts with fly spike-ins

First, reads are assigned as originating from either human, fly, yeast or ERCC, based on alignment to hybrid genome. Then, human raw counts are divided by the sum of all fly spike-in raw counts. Since both human and fly RNAs are 4sU-labeled, this normalization to fly spike-ins reconstructs the fraction of 4sU-labeled human RNA at each time point.

### Spike-in RNA usage clarification

Fly spike-ins were used to normalize human counts for all time points (0.5-hour to 24-hour) to assist with transcription rate and absolute half-life calculations. Yeast spike-ins are only used as quality control for contamination in the pull-downs as presented in Fig. S1. ERCC spike-ins are only used to control for sequencing quality as presented in Fig. S1. First, it directly estimates sequencing error magnitude, excluding biological variation. Second, it shows the capacity of 3′-end QuantSeq to reconstruct molar concentrations. Note that the steady-state sample is not normalized with spike-ins and is separate from the 24-hour time point sample.

### RNA dynamics in 4sU incorporation experiment

Let Y(t) be human normalized counts at time t, $Y_{SS}$ be human normalized counts at steady-state, $\alpha$ transcription rate, and $\beta$ RNA degradation rate. Then, Y(t) changes over time t as follows:

$$\frac{d\mathrm{Y(t)}}{dt} = \alpha - \beta\mathrm{Y(t)}. \tag{1}$$

At steady − state $\left(\frac{dY}{dt} = 0\right)$ :

$$\alpha = \beta\mathrm{Y}_{ss}. \tag{2}$$

Then, 4sU incorporation kinetics is a solution to above equation, when Y(t)=0 at time t = 0:

$$\mathrm{Y(t)} = \mathrm{Y}_{ss}\left(1 - e^{-\beta t}\right). \tag{3}$$

In general, cell division dilutes 4sU labeled transcripts. This effect can be accounted for by using $\beta + \beta_{growth}$ instead of just $\beta$. However, $\beta_{growth}$ contribution is absent for our experiment since neurons are post-mitotic. Then, at early time points (t → 0):

$$\frac{d\mathrm{Y(t)}}{dt}\bigg|_{\mathrm{t}\to 0} = \beta\mathrm{Y}_{ss}. \tag{4}$$

Note that $\beta\mathrm{Y}_{ss}$ is equal to transcription rate $\alpha$. This means that slope of Y(t) around t = 0 measures transcription rate. This property is used in the analysis described below.

### Transcription rate and half-life measurements

The transcription rate was estimated from 0.5-hour and 1-hour time points for WT neurons and only from 0.5-hour time point for RTT neurons. This estimate assumes that RNA degradation is negligible for most genes at time points before 1 hour (see Supplementary Data 1 for previously published mRNA half-life measurements using different cell models and techniques). First, human normalized counts at 1-hour time point are divided by 2 to create a new approximate replicate at 0.5-hour time point. Division by 2 is done to account for a twice longer period of transcription in a 1-hour time point sample. To clarify, the addition of an approximate replicate was motivated by the reduction in the standard error of log₂FC estimated with *DESeq2*. To compare the transcription rate between cell types using *DESeq2*, human normalized counts are further quantile normalized between replicates of the same cell type with *normalize.quantiles* (*preprocessCore* package)[57,58]. Half-life was estimated in 2 separate ways: fit of the 4sU saturation curve and the ratio of steady-state to transcription rate. For the 4sU saturation curve method, the half-life is estimated in a 2-step procedure. First, normalized counts are fit with *nls* (nonlinear least squares from *stats* package) to approximate the true number of counts Y at each timepoint. Then, in a second pass, normalized counts are fit again with *nls*, but now correcting for the increase in variance using weights set as 1/Y. Confidence intervals are estimated with *confint* function (*stats* package). For the ratio method, the half-life is estimated with *DESeq2* using raw human counts from 0.5-hour, 1-hour and steady-state

samples (design = ~ assay). The assay is a 2-level factor one for transcription rate and one for steady-state. 0.5-hour and 1-hour samples correspond to the transcription rate.

### A caveat of using a pseudo-replicate

In brief, the transcription rate measurement aims to quantify a fold-change between genotypes from the replicates. A true replicate incorporates all sources of variation, including pipetting or plate handling. All time points of our RATEseq experiment are grown on the same multi-well plate. Thus, 0.5 h & 1 h samples account only for inter-well but not for inter-plate variation. Such a replicate is a pseudo-replicate. The inclusion of a pseudo-replicate may artificially reduce variance and introduce bias.

### The shift in average mRNA half-life between genotypes

To describe the global absolute mRNA half-life shift, we filtered out unreliable genes from the saturation curve method. First, both gene transcription rate and steady-state should be above the bottom 10% quantile. Second, the ratio of 50% confidence interval and estimate of the half-life should be below 0.75. Then, the mean and median half-life was calculated from the filtered set of genes.

### Processing of mouse datasets

Mouse data for whole-cell, nuclear and chromatin RNA-seq was downloaded from GSE128178. Mouse *Mecp2* ChIP-seq was downloaded from GSE139509. Differential expression analysis for nuclear and chromatin RNA-seq was downloaded from the supplementary materials of the Boxer et al. study[9]. The abundance of 3′UTR isoforms for all samples is estimated using the QAPA standard pipeline[59]. Half-life was estimated as a ratio between whole-cell counts and nuclear or chromatin counts using the interaction term approach in *DESeq2* (design = ~ celltype + batch + assay + celltype:assay). Assay factor encodes whole-cell vs nuclear or chromatin samples. Coefficient of the celltype:assay term is used to measure the log2 fold-change in half-life between cell types. Before the *DESeq2* run, we filter out genes with a sum of counts in replicates less than 20 in a pair of compared cell types.

### Random forest prediction of up and down-regulated genes in transcription rates and mRNA half-life

Fold-changes in human half-life in log scale $\log_2 FC_{HL}$ between cell types A and B were estimated as follows:

$$\log_2 FC_{HL} = \log_2 HL_B - \log_2 HL_A. \qquad (5)$$

$$Z_{FC} = |\log_2 FC_{HL}| / \sqrt{lfcSE_B^2 + lfcSE_A^2}$$

There, $\log_2 HL_A$ and $lfcSE_A$ were average and standard error of the half-life in cell type A, estimated by *DESeq2* as a ratio from steady-state and transcription rate replicates. Z-score of a fold-change $Z_{FC}$ was used as a measure of accuracy.

Features of the classifier are frequencies of k-mers in 3′UTR, coding sequence or gene-body, calculated using *oligonucleotide-Frequency* (*Biostrings* package)[60]. In addition, the methylation status of CA and CG in the gene-body was added for mouse analysis from GSE139509. Predicted variable denotes genes that are either up or down-regulated in transcription rate or half-life. Thresholds for the human data were:

1. $TR_{up}$: $\log_2 FC_{TR} > 1$ & padj < 0.1
2. $TR_{down}$: $\log_2 FC_{TR} < -1$ & padj < 0.1
3. $HL_{up}$: $\log_2 FC_{HL} > 1$ & $Z_{FC} >$ median($Z_{FC}$)
4. $HL_{down}$: $\log_2 FC_{HL} < -1$ & $Z_{FC} >$ median($Z_{FC}$)

The half-life for the mouse was either from nuclear or chromatin. Thresholds for the mouse data were:

1. $TR_{up}$: $\log_2 FC_{TR} > 0.1$ & FDR < 0.1
2. $TR_{down}$: $\log_2 FC_{TR} < -0.1$ & FDR < 0.1
3. $TR_{not}$: $|\log_2 FC_{TR}| < 0.1$ or FDR > 0.1
4. $HL_{up}$: $\log_2 FC_{HL} > 1$ & pvalue < quantile(pvalue, 0.2)
5. $HL_{down}$: $\log_2 FC_{HL} < -1$ & pvalue < quantile(pvalue, 0.2)

Data was split into 75% and 25% for training and test sets. The classifier is trained using *randomForest* (*randomForest* package)[61]. Precision-recall and receiver operator characteristic curves were obtained with *evalmod* (*precrec* package)[62].

### Transite analysis of miRNAs and RBPs

Genes were split into up- or down-regulated according to their transcription rate and half-life fold-change. Then, we performed multiple comparisons of 3′UTR sequences between groups of genes using *run_kmer_tsma* (*transite* package)[63]. Groups of compared genes:

1. Foreground: $TR_{down}$ and $HL_{up}$ Background: $TR_{down}$
2. Foreground: $TR_{up}$ and $HL_{down}$ Background: $TR_{up}$
3. Foreground: $TR_{not}$ and $HL_{down}$ Background: $TR_{not}$

These comparisons were performed for all transite RNA binding protein motifs and for TargetScan seed sequences[32]. TargetScan analysis includes 439 human miRNA-seed sequences with family conservation scores of 0,1,2 that were selected from miR_Family_Info.txt (TargetScan website). Definitions of up- and down-regulated genes:

1. $TR_{up}$: $\log_2 FC_{TR} > 0.5$ & padj<0.1
2. $TR_{down}$: $\log_2 FC_{TR} < -0.5$ & padj < 0.1
3. $TR_{not}$: $|\log_2 FC_{TR}| < 0.5$ or padj > 0.1
4. $HL_{up}$: $\log_2 FC_{HL} > 1$ & $Z_{FC} >$ median($Z_{FC}$)
5. $HL_{down}$: $\log_2 FC_{HL} < -1$ & $Z_{FC} >$ median($Z_{FC}$)

The results of *transite* analysis were further processed with a custom script for multiple hypothesis correction. Motifs with a low number of sites detected in both background and foreground were removed from the analysis. A separate threshold for the number of sites was chosen for each transite analysis. A threshold was determined from a requirement for *p*-values distribution to be unimodal and enriching at $p = 0$. The distribution of *p*-values with unfiltered sites is bimodal with peaks at both $p = 0$ and $p = 1$.

### Reporting summary

Further information on research design is available in the Nature Portfolio Reporting Summary linked to this article.

## Data availability

The data that support this study are available from the corresponding author upon reasonable request. The original sequencing data generated in the course of this study can be accessed from GEO using the access number GSE191168. The mouse data reanalyzed in this study can be downloaded from GSE128178 (whole-cell, nuclear and chromatin RNA-seq) and GSE139509 for MECP2 ChIP-seq. The external data used on the heatmap comparison of neurons can be downloaded from GSE38805 (neocortex) and GSE86985 (iPSC-derived dual-SMAD neurons) and GSE112732 (iPSC-derived dual-SMAD NGN2 neurons). Source data are provided with this paper.

## Code availability

The computational code used in this manuscript is available at Github – https://github.com/JellisLab/stabilome-rett [64] under https://doi.org/10.5281/zenodo.7537735.

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

## Acknowledgements

This study was funded by grants from the Canadian Institutes of Health Research (CIHR; PJT-148746, PJT-168905, and ERARE Team ERT 161303 to J.E.); the Canada First Research Excellence Fund (Medicine by Design Cycle I: J.E.); the Col. Harland Sanders Rett Syndrome Research Fund at the University of Toronto (J.E.); the Ontario Brain Institute (POND Network: J.E.); and John Evans Leaders Fund/Canada Foundation for Innovation (JELF/CFI: J.E); Canada Research Chairs Program (M.D.W. and J.E.); Early Researcher Award from the Ontario Ministry of Research and Innovation (M.D.W.); Genome Canada Disruptive Innovation in Genomics Grant (M.D.W to support K.E.Y.); NIH grant R35GM128680 and the University of Colorado RNA Bioscience Initiative (O.R.); David Steven Cant Scholarship (M.M.). We thank the Centre for Applied Genomics (TCAG) at SickKids for RNA sequencing, and Brian Kalish for comments on the manuscript. Some figure panels were created with BioRender.com.

## Author contributions

Conceptualization, D.C.R., M.M., A.N., O.S.R., and J.E.; Investigation, D.C.R., M.M., A.N., K.Y., W.W., A.P., and J.L.; Software, M.M., and A.N.; Writing—original draft, D.C.R., M.M., and J.E.; Writing—Review and Editing, D.C.R., M.M., A.N., K.Y., P.P., O.S.R., M.D.W., and J.E.; Visualization, D.C.R., and M.M.; Supervision and Funding Acquisition, O.S.R., M.D.W., and J.E.

## Competing interests

The authors declare no competing interests.
