## [Peer Review File · Nature Communications]

Buffering of transcription rate by mRNA half-life is a conserved feature of Rett syndrome modelsREVIEWER COMMENTS

Reviewer #1 (Remarks to the Author):

This manuscript from Rodrigues et al. challenges the previously established notion that transcription rate changes associated with MECP2 dysfunction in Rett syndrome explain alterations in steady-state mRNA levels. Evidence preceding this work already demonstrated a support to an opposing notion that transcription rate changes might be buffered by compensatory post-transcriptional mechanisms. Following this lead, authors presented a compelling evidence that transcription rate changes in Rett syndrome is buffered by compensatory mRNA half-life changes, which minimizes alterations in steady-state mRNA levels. To this end, authors demonstrated that combinations of 3-dinucleotides including previously identified CA/CG motifs are better predictors of transcription rate changes than the gene-body lengths. Moreover, they identified two exclusive gene sets where one set shows mRNA half-life changes with no apparent changes in transcription rate and the other set shows compensation of transcription rate changes by mRNA half-life regulation resulting in global increase in mRNA half-lives. Finally, to present a mechanistic view as to how transcription rate changes are buffered, they demonstrated that in MECP2-null neurons RNA Binding Proteins (RBPs) binds to 3'UTRs and other distinct sites in gene-bodies to post-transcriptionally regulate RNA half-lives. Overall, this study presents a new viewpoint of MECP2-associated gene dysregulations in Rett syndrome and is a potentially impactful study that may carry forward our understanding of Rett syndrome etiology. The manuscript is well-written and the figures are well-organized. There are only minor suggestions to improve the manuscript and support their conclusion.

1. Authors should include a supplementary figure with a simple drawing of cortical neuron differentiation protocol and a panel of immune-stained cortical neurons with a few known cortical neuron markers.
2. A rationale of how authors choose the time points for steady-state levels (24-hour) and transcription rate (the slope of 0.5-1 hour) is missing in the paper. An explanatory sentence (or two) should be added to ease the understanding of their manuscript for people out of the field.
3. Although in their introduction where they summarize their findings authors state that transcription rate changes are best predicted by combinations of 3-dinucleotide frequencies that include the previously identified CA/CG motifs, in the Result section where they described Figure 2B, they stated that the removal of CA/CG from the model had no negative effect on prediction accuracy. This looks like a discrepancy between the statements. Please revise these sections to make statements consistent.
4. Similarly, the statement where authors described Figure 2D-E, directly quoting "The top four combinations of three dinucleotides were highly similar between human and mouse, and included AT as enriched along with CA and CG and other dinucleotides." seems not to fit the data shown in the figure. It is difficult to see high similarity between human and mouse 3-dinucleotide combinations and contrary to the statement, AT dinucleotide does not seem to be enriched in neither human nor mouse and as a matter of fact that mouse does not contain CA dinucleotide at all. Is this the result of comparing the data from immature (human) and mature (mouse) neurons? In any case, the result presented in Figure 2D-E seems to be overstated in the text and needs to be revised better reflecting the results presented in the figures.
5. In line 194-196, authors concluded that although human and mouse RTT models show a conserved relation of half-life and transcription rate on steady-state gene expression, there was a minimal overlap in the identities of genes dysregulated in each species. This sounds like an interesting and important conclusion and speaks against the idea that mouse models, at least for Rett syndrome, recapitulates human-specific characteristic of disease etiology. It would be worthwhile to add a small part in the Discussion on similar differences previously shown between human and animal subjects citing relevant literature.

Reviewer #2 (Remarks to the Author):

This study from Rodrigues et al. addresses an outstanding question in the Rett syndrome (RTT) research field, and their findings have broad implications to the study of gene regulation in general. RTT is a childhood neurological disorder caused by loss-of-function mutations in the gene encoding methyl-CpG binding protein 2 (MeCP2). MeCP2 has been shown to bind broadly across the genome to methylated cytosines, and the current prevailing hypothesis about MeCP2 function is that it mediates transcriptional repression. However, transcriptome profiling in animal models of RTT (as well as in human cells and brain tissues) have revealed subtle, yet numerous changes in gene expression, with approximately equal up- and down-regulated genes. Given the severity of the disorder, it has been challenging to understand how the gene expression changes associated with RTT are modest in magnitude. In the current study, the authors built on previous work suggesting that these modest changes detected by standard steady-state RNA-seq may be confounded by RNAs sequenced from different cellular compartments, cell types and changes in RNA stability. Using human RTT patient-derived neurons, the authors explored this hypothesis in great detail, uncovering large differences in both RNA synthesis and decay, and presented evidence that changes in both processes may explain the subtle changes in steady-state RNA levels as detected by standard RNA-seq. The authors subsequently employed a machine learning approach to investigate the association between gene expression changes and gene-body DNA sequence and found distinct dinucleotide sequences that were better predictors of transcriptional changes than cytosines in the CA or CG context (which can be methylated and are thus thought to be MeCP2's canonical binding targets). Finally, the authors studied how these broad post-transcriptional changes could be occurring, and provide evidence that both miRNA and RNA binding protein (RBP) function may underlie the changes in RNA synthesis and decay in RTT.

The findings from this study are an important contribution to the RTT/MeCP2 research field and may help explain the perplexing results from steady-state RNA-seq experiments that are commonly used to infer gene regulation. The study employs sophisticated techniques in a disease-relevant model of RTT, providing the opportunity for mechanistic insights with a high degree of experimental controls. There are multiple questions/points concerning the methodology, experimental design, and interpretation that should be appropriately addressed prior to publishing in Nature Communications.

Major points:

1. This study hinges on, to a large extent, the ability to properly measure RNA synthesis and degradation rates. To measure RNA half-life, it seems like two different methods were employed to estimate the degradation rates: the first is by using nonlinear regression (as was done in the original RATE-seq paper), and the second is by calculating the ratio between the steady state and transcription rate. It is unclear which one is used where, as the manuscript seems to jump back and forth. Furthermore, a justification for using the second approach over the first is lacking – why is this a valid way to calculate half-life? Furthermore, although the actinomycin D experiment was a nice validation, it is unclear how the half-lives were calculated with this method.

2. To measure transcription rates, the method applied in this study assumes no labeled transcripts would degrade over the first hour of labeling. This is an assumption and there is no orthogonal evidence suggesting that this assumption is valid. The authors describe in the results section that “the transcription rate was measured from 0.5- and 1-hour time-points, by calculating the number of newly synthesized mRNAs in 1 hour.” However, the methods section notes that the 1h timepoint for the MECP2-null line was omitted from both replicates due to low differentiation. If that is the case, how were the transcription rate calculations performed for the RTT samples? Additionally, the methods section lacks several other critical details about how the transcription rate was calculated: the fly spike-in-normalized counts at the 0.5h and 1h timepoints were quantile normalized? Was the difference between these two values divided by the time in between them (0.5h) to get the transcription rate? Why were the counts at 1 hour divided by 2 to generate a pseudoreplicate –

weren't there enough replicates at the 0.5h timepoint to begin with?

3. To overcome challenges associated with estimating transcript half-life with the experimental setup described in the manuscript, a 4sU pulse-chase sequencing experiment would be more appropriate to characterize the transcript half-life and support the overall conclusion.

4. No information was provided about the number of biological and technical replicates. Although it is evident that the RTT patient-derived neurons came from an individual patient (and thus there is one biological replicate), the number of technical replicates at each timepoint in the RATE-seq experiment, as well as all the other experiments, are not provided in the text or the figure legends. In addition, information about how RATE-seq measures RNA dynamics is missing, leaving it difficult to assess how it estimates synthesis and decay rates by modeling RNA synthesis as a constant and RNA abundance with an exponential decay model.

5. This study used multiple spike-in controls, but sufficient details about each and how the spike-in data were analyzed are lacking. By piecing together information from supplementary figure legends, the methods, and the main text, it sounds like the fly spike-in was used to normalize for differences in pulldown efficiency, the yeast spike-in was used as negative control (not used to scale the counts), and the ERCC spike-in was used to compare RNA input for library prep (but not used to scale the sequencing counts). Very little rationale was provided in the manuscript, making it challenging to assess the data quality and conclusions.

6. While the global decrease in miRNAs is interesting, one may argue that this is an artifact of the spike-in strategy. If the spike-ins aren't properly calibrated across each condition, the data will be mistakenly scaled up or down to give the illusion of a global increase or decrease. Additional controls and validation would be needed to support the conclusions made in the manuscript about miRNAs.

Minor points:

1. Although differences across subcellular compartments in steady-state RNA-seq from the Boxer et al dataset imply the existence of post-transcriptional mechanisms affecting RNA dynamics in RTT, this is compatible with, but not equivalent, to the authors' findings about transcription rate and half-life. In another words, single-timepoint transcriptome profiling from the nucleus or from chromatin-associated RNA is not a sufficient proxy for transcription rate and single-timepoint whole-cell RNA-seq is not a sufficient proxy for the steady-state. The way this is discussed and the conclusions made in the text should be changed (e.g. lines 118-120, the labels in Figure 1F-H should name the different subcellular compartments, which are not the same as rates and steady-state). Furthermore, the analysis presented in the section titled "Half-life shifts and transcription buffering are conserved in Mecp2 mouse models" should be changed to reflect differences in relative RNA abundance between genotypes across subcellular compartments – this is not directly equivalent to half-life changes and changes in transcription rate.

2. Although the authors have previously published the generation and characterization of their isogenic RTT iPSC-derived neurons, it is critical to include evidence showing the differentiation of human cortical neurons and the expression of MeCP2 between lines in the current study.

3. In Figure 3, the source of the data displayed in panels F, G and H is unclear – is this a comparison between the Mecp2-null mice or the R306C mice, or were they merged? In Figure 3I these two lines are shown separately.

4. Based on what is presented in Supplementary Figure 1, the 4sU-labeled fly spike-in RNA seems to be much more abundant than 4sU-labeled pulled-down human RNA. Will the excessive spike-in RNA lead to insufficient sequencing depth and inaccurate profiling? Dot blots would be more appropriate to show the input.

5. Critical information is missing from Figure 5 C and D. Do those boxplots show total fold-changes? Before and after scaling to spike-ins?

6. Line 251-252: What about miRNA-binding sites in genes that only display changes in transcription rate and in genes that display buffering in the opposite direction (decreased TR and increased HL)?

7. Supplementary Fig 1C, the genotype labels seem to be switched (the NEUwt should be over the right gel and NEUrtt should go over the left gel, given that the authors mentioned not being able to obtain the 1h timepoint in the RTT condition).

Reviewer #1:

This manuscript from Rodrigues et al. challenges the previously established notion that transcription rate changes associated with MECP2 dysfunction in Rett syndrome explain alterations in steady-state mRNA levels. Evidence preceding this work already demonstrated a support to an opposing notion that transcription rate changes might be buffered by compensatory post-transcriptional mechanisms. Following this lead, authors presented a compelling evidence that transcription rate changes in Rett syndrome is buffered by compensatory mRNA half-life changes, which minimizes alterations in steady-state mRNA levels. To this end, authors demonstrated that combinations of 3-dinucleotides including previously identified CA/CG motifs are better predictors of transcription rate changes than the gene-body lengths. Moreover, they identified two exclusive gene sets where one set shows mRNA half-life changes with no apparent changes in transcription rate and the other set shows compensation of transcription rate changes by mRNA half-life regulation resulting in global increase in mRNA half-lives. Finally, to present a mechanistic view as to how transcription rate changes are buffered, they demonstrated that in MECP2-null neurons RNA Binding Proteins (RBPs) binds to 3'UTRs and other distinct sites in gene-bodies to post-transcriptionally regulate RNA half-lives. Overall, this study presents a new viewpoint of MECP2-associated gene dysregulations in Rett syndrome and is a potentially impactful study that may carry forward our understanding of Rett syndrome etiology. The manuscript is well-written and the figures are well-organized. There are only minor suggestions to improve the manuscript and support their conclusion.

Thank you for the positive comments. We have inserted additional figure panels, rewritten the first section of the results for more clarity about the experimental design and methods, removed the minus CA/CG classification result and the “enriched” three dinucleotide combination conclusion, and discuss the minimal overlap of specific genes that are differentially expressed across models.

1. Authors should include a supplementary figure with a simple drawing of cortical neuron differentiation protocol and a panel of immune-stained cortical neurons with a few known cortical neuron markers.

Following the reviewer's suggestion, we added a schematic in new Fig S1A depicting the major steps of our differentiation protocol.

We also performed a comparative analysis of the steady-state mRNA levels of a large panel of neural marker genes (48 genes) between our neurons and other previously published transcriptomics studies on pluripotent stem cell-derived neurons and human fetal neocortex (new Fig 1C). This analysis validates our differentiation protocol on the actual cell batches used for RATESeq, and the transcriptomic profile is more comprehensive than verification by immune-staining with a few known cortical neuron markers on a different batch of cells.

2. A rationale of how authors choose the time points for steady-state levels (24-hour) and transcription rate (the slope of 0.5-1 hour) is missing in the paper. An explanatory sentence (or two) should be added to ease the understanding of their manuscript for people out of the field.

We have rewritten the first section of the Results and revised the Methods section to better explain the rationale behind those choices. In brief, steady-state was measured at the 24-hour time point in order to have all RNA samples quantified in the presence of 4sU, accounting for unexpected deleterious effects of 4sU treatment. For transcription rate, 0.5 and 1 hour were chosen as at these early time-points, degradation of newly transcribed mRNA in the nucleus is negligible for most genes. We also compiled multiple studies measuring mRNA half-life in different cell types using different methodologies (new Figure 3B and Supplementary table 3). This analysis indicates that for the vast majority of genes, mRNA half-life is greater than 2.5 hours, therefore well above the time points of 0.5 and 1 hour we chose to measure transcription rates.

3. Although in their introduction where they summarize their findings authors state that transcription rate changes are best predicted by combinations of 3-dinucleotide frequencies that include the previously identified CA/CG motifs, in the Result section where they described Figure 2B, they stated that the removal of CA/CG from the model had no negative effect on prediction accuracy. This looks like a discrepancy between the statements. Please revise these sections to make statements consistent.

After further consideration, we concluded that the analysis was not informative and confusingly presented. We thank the reviewer for spotting this inconsistency. For clarity, we decided to refine the text and Figures 2D-E by removing all “minus CA/CG” models.

To reiterate, the goal of the analysis was to probe the predictive power of a classifier without the CA and CG dinucleotides. However, this approach suffers from a mild correlation between dinucleotides. In short, the distribution of a dinucleotide X frequency across genes can resemble the distribution for another dinucleotide Y. Therefore, the “minus CA/CG” model still contains the information of the CA and CG dinucleotides through correlation and this in turn inflates the prediction accuracy of the “minus CA/CG” model. We already proposed a solution to this issue in our dinucleotide combination model in Figure 2D. There, we tried to identify the minimal set of dinucleotides that can reach the prediction power of the full model. Testing combinations of 2 or 3 dinucleotides in isolation is a cleaner way to probe individual dinucleotides’ contributions without being affected by correlations with other dinucleotides.

4. Similarly, the statement where authors described Figure 2D-E, directly quoting “The top four combinations of three dinucleotides were highly similar between human and mouse, and included AT as enriched along with CA and CG and other dinucleotides.” seems not to fit the data shown in the figure. It is difficult to see high similarity between human and mouse 3-dinucleotide combinations and contrary to the statement, AT dinucleotide does not seem to be enriched in neither human nor mouse and as a matter of fact that mouse does not contain CA dinucleotide at all. Is this the result of comparing the data from immature (human) and mature (mouse) neurons? In any case, the result presented in Figure 2D-E seems to be overstated in the text and needs to be revised better reflecting the results presented in the figures.

As for the previous point, we modified the text and figures to more precisely describe the results of our predictive models. The main point of this new reiteration is that in both species the model indicates that MECP2 functions through a combination of 3 dinucleotides even though the combinations of 3 dinucleotides are not necessarily the same between human and mouse. This difference could be due to a species-specific MECP2 binding capacity or the different developmental stages between human and mouse samples analyzed.

5. In line 194-196, authors concluded that although human and mouse RTT models show a conserved relation of half-life and transcription rate on steady-state gene expression, there was a minimal overlap in the identities of genes dysregulated in each species. This sounds like an interesting and important conclusion and speaks against the idea that mouse models, at least for Rett syndrome, recapitulates human-specific characteristic of disease etiology. It would be worthwhile to add a small part in the Discussion on similar differences previously shown between human and animal subjects citing relevant literature.

That is an interesting point raised by the reviewer and it had already been observed in the Boxer et al study. We now cite a previous study by the Tropea group that showed a high level of disagreement between transcriptomic studies between mouse RTT models which agrees with our observation. However, there seems to be a core group of genes consistently dysregulated across all models. The most important point made is that despite this variability, the mRNA decay mechanisms seem to work with considerable flexibility in order to buffer different groups of mRNAs.

We added a new paragraph to the discussion section to raise these points.

Reviewer #2 (Remarks to the Author):

This study from Rodrigues et al. addresses an outstanding question in the Rett syndrome (RTT) research field, and their findings have broad implications to the study of gene regulation in general. RTT is a childhood neurological disorder caused by loss-of-function mutations in the gene encoding methyl-CpG binding protein 2 (MeCP2). MeCP2 has been shown to bind broadly across the genome to methylated cytosines, and the current prevailing hypothesis about MeCP2 function is that it mediates transcriptional repression. However, transcriptome profiling in animal models of RTT (as well as in human cells and brain tissues) have revealed subtle, yet numerous changes in gene expression, with approximately equal up- and down-regulated genes. Given the severity of the disorder, it has been challenging to understand how the gene expression changes associated with RTT are modest in magnitude. In the current study, the authors built on previous work suggesting that these modest changes detected by standard steady-state RNA-seq may be confounded by RNAs sequenced from different cellular compartments, cell types and changes in RNA stability. Using human RTT patient-derived neurons, the authors explored this hypothesis in great detail, uncovering large differences in both RNA synthesis and decay, and presented evidence that changes in both processes may explain the subtle changes in steady-state RNA levels as detected by standard RNA-seq. The authors subsequently employed a machine learning approach to investigate the association between gene expression changes and gene-body DNA sequence and found distinct dinucleotide sequences that were better predictors of transcriptional changes than cytosines in the CA

or CG context (which can be methylated and are thus thought to be MeCP2's canonical binding targets). Finally, the authors studied how these broad post-transcriptional changes could be occurring, and provide evidence that both miRNA and RNA binding protein (RBP) function may underlie the changes in RNA synthesis and decay in RTT.

The findings from this study are an important contribution to the RTT/MeCP2 research field and may help explain the perplexing results from steady-state RNA-seq experiments that are commonly used to infer gene regulation. The study employs sophisticated techniques in a disease-relevant model of RTT, providing the opportunity for mechanistic insights with a high degree of experimental controls. There are multiple questions/points concerning the methodology, experimental design, and interpretation that should be appropriately addressed prior to publishing in Nature Communications.

We agree that the RATESeq methods and experimental design used in Figure 1 and S1 were inadequately described in the Results section. We have extensively rewritten this important first section of the Results to walk more slowly through each panel and all the supplemental material. "Steady-state" is now more clearly used in the restricted context of an equilibrium between transcription rate and half-life in the whole-cell only.

For clarity, we corrected Fig 1A summarizing the RATESeq method to show that the steady-state 24 hour sample is different from the 24 hour RATESeq processed sample, and that this was used to calculate the steady-state for the Ratio method. We also indicate in Fig 1A that the absolute half-life is calculated from the 4sU saturation curve, the half-life fold changes are calculated using the Ratio method, and the Transcription Rate is deduced from the 0.5 and 1 hour time points.

Specific revisions are explained at greater length in the points below.

Major points:

1. This study hinges on, to a large extent, the ability to properly measure RNA synthesis and degradation rates. To measure RNA half-life, it seems like two different methods were employed to estimate the degradation rates: the first is by using nonlinear regression (as was done in the original RATE-seq paper), and the second is by calculating the ratio between the steady state and transcription rate. It is unclear which one is used where, as the manuscript seems to jump back and forth. Furthermore, a justification for using the second approach over the first is lacking – why is this a valid way to calculate half-life? Furthermore, although the actinomycin D experiment was a nice validation, it is unclear how the half-lives were calculated with this method.

We thank the reviewer for noticing the lack of clarity in how we described the half-life experiments. We edited the Methods sections to better explain the rationale and approaches used in each measurement. We also modified the Results section to clarify the use of distinct methods to calculate half-life.

We also added more information to the Results, Methods, and figure legend sections regarding the Actinomycin D experiment.

In brief, nonlinear regression of the 4sU saturation curves quantifies half-life in hours or in absolute scale, while the ratio method outputs the relative change in half-life. Both methods output similar half-lives for genes with well-measured saturation curves as you can see in the figure S1M below.

However, the saturation curve method struggles to quantify genes with relatively low transcription rates (transcription rate counts [TR] in the figure S1K below).

After we describe the limitation of the saturation curve method in Figs S1K-M, it is subsequently only used to determine the median half-life values in Figure 3A which compares the absolute half-life (WT = 2.5 hours, RTT = 3.0 hours) in the human neurons. We now describe the detailed method used for this calculation in the methods and also add to the text the corresponding mean absolute half-life values (WT = 2.9 hours, RTT = 4.6 hours) to further support this difference. All other relative half-life measurements in Figures 2 and 3 consistently use the ratio method to increase the number of quantified genes.

2. To measure transcription rates, the method applied in this study assumes no labeled transcripts would degrade over the first hour of labeling. This is an assumption and there is no orthogonal evidence suggesting that this assumption is valid. The authors describe in the results section that “the transcription rate was measured from 0.5- and 1-hour time-points, by calculating the number of newly synthesized mRNAs in 1 hour.” However, the methods section notes that the 1h timepoint for the MECP2-null line was omitted from both replicates due to low differentiation. If that is the case, how were the transcription rate calculations performed for the RTT samples? Additionally, the methods section lacks several other critical details about how the transcription rate was calculated: the fly spike-in-normalized counts at the 0.5h and 1h timepoints were quantile normalized? Was the difference between these two values divided by the time in between them (0.5h) to get the transcription rate? Why were the counts at 1 hour divided by 2 to generate a pseudoreplicate – weren’t there enough replicates at the 0.5h timepoint to begin with?

The reviewer is right that we assumed no degradation of labeled RNAs at the 0.5 and the pseudoreplicate 1 hour time points. This assumption was based on a few factors: 1) Our absolute half-life measurement using the saturation curve method in Figure 3A indicates that the median mRNA half-life in both cells is 2.5 hours or greater and therefore well above 30 mins. 2) the literature on mRNA half-life measurements indicates that the median mRNA half-life is routinely at or above 2.5 hours. Please see below a table where we compile multiple studies using different measurement techniques, most of which agree that mRNA half-life is above 2.5 hours, except when very short label times or the “near triploid” K562 erythroleukemia cell line is used. We decided to include the table below as a new supplementary table 3 to the manuscript. We also plotted a graph, shown below, that summarizes the information from the table (new Fig 3B). 3) Finally, from a technical perspective, in a pilot experiment on iPSC-derived neurons we concluded that shorter 4sU pulses (less than 0.5 hour) did not yield enough labeled RNA to undergo the entire protocol of biotinylation, pull-down, library prep, and sequencing.

The low yield of MECP2-null neurons forced us to drop the 1 hour time point for this line. Therefore, the transcription rate for the MECP2-null samples was calculated using the 0.5 hour replicates. We thank the reviewer for spotting that and we have modified the methods section and results to incorporate this critical information. In short, human raw counts are normalized with fly spike-in RNA. Then, only the human normalized counts of 0.5 and 1 hour time points are quantile normalized. Then, we reasoned that more replicates would enhance the log₂ fold-change measurements. Indeed, the addition of a pseudo-replicate reduced the standard error in the measured log₂ fold-changes (please see below Revision Fig 2).

Method	Cell line	Median (hrs)	First Author	Journal	Year
Flavipiridol 8hrs	OCI-Ly3	8	Lam	Gen Bio	2001
ActD 3hrs	HepG2	10	Yang	Gen Res	2003
ActD 4hrs	hiPSC HFF	8.6 9.2	Neff	Gen Res	2012
ActD 6hrs	HeLa	4.6	Sharma	Mol Cell	2016
ActD 8hrs	mESC	7.1	Sharova	DNA Res	2009
ActD 8hrs	mESC	8	Zheng	Stem Cell Rep	2016
BrU 24hrs 150uM pulse	HeLa	7	Tani	Gen Res	2012
4sU 5min 500uM	K562	0.8	Schwalb	Science TTseq	2016
4sU 10min 500uM	Mouse DC	2.3	Rabani	Cell	2014
4sU 10min 150uM	Mouse BMDC	0.5	Rabani	Nat Biotech	2011
4sU 1hrs MEFs 1000uM 4hrs K562 100uM	MEFs K562	2.7 1.6	Schofield	Nat Methods TimelapseSeq	2018
4sU 1hrs 200uM	Mouse NIH3T3	4.9	Dolken	RNA	2008
4sU 1hrs	Human B Murine fibroblast	5.2 4.6	Friedel	Nuc Acid Res	2009
4sU 2hrs 400uM	Mouse NIH3T3	9.9	Schwanhausser	Nat	2011
4sU 24hrs 100uM pulse exchange every 3hrs	mESC	3.9	Herzog	Nat Methods SlamSeq	2018

Revision Fig 1.

Revision Fig 2.

3. To overcome challenges associated with estimating transcript half-life with the experimental setup described in the manuscript, a 4sU pulse-chase sequencing experiment would be more appropriate to characterize the transcript half-life and support the overall conclusion.

We agree with the reviewer that 4sU pulse-chase is an accurate orthogonal approach to measure half-life (HL). Analysis of a linear fit is indeed easier, but it has recently been shown that there is high reproducibility comparing a RATESeq +spike-in enrichment protocol in parallel to pulse-chase approaches for measuring half-life (e.g., Boileau E. et al., PMID: 34228787). We support our conclusions with orthogonal primary validations (EU incorporation for TR and Actinomycin D for HLs), and we now cite the Boileau paper suggesting that secondary validation using a pulse-chase approach would further support the half-life conclusions.

4. No information was provided about the number of biological and technical replicates. Although it is evident that the RTT patient-derived neurons came from an individual patient (and thus there is one biological replicate), the number of technical replicates at each timepoint in the RATE-seq experiment, as well as all the other experiments, are not provided in the text or the figure legends. In addition, information about how RATE-seq measures RNA dynamics is missing, leaving it difficult to assess how it estimates synthesis and decay rates by modeling RNA synthesis as a constant and RNA abundance with an exponential decay model.

We apologize for missing this information. We added all details related to replicate numbers to the figures, figure legends, and methods. We also added the information related to the RNA dynamics to the methods section.

5. This study used multiple spike-in controls, but sufficient details about each and how the spike-in data were analyzed are lacking. By piecing together information from supplementary figure legends, the methods, and the main text, it sounds like the fly spike-in was used to normalize for differences in pulldown efficiency, the yeast spike-in was used as negative control (not used to scale the counts), and the ERCC spike-in was used to compare RNA input for library prep (but not used to scale the sequencing counts). Very little rationale was provided in the manuscript, making it challenging to assess the data quality and conclusions.

Once again, we apologize to the reviewer for the lack of critical information. We added all the information regarding the use of spike-ins to the methods section and two references describing their use in RATESeq (Lugowski A. et al, and Boileau E. et al). We also modified Fig 1 to add information on the use of each spike-in RNA. In brief, the spikes were used for: 4sU labeled Fly= normalization of human counts; unlabeled Yeast= background contamination (unlabeled RNA present in the pull-down fraction); ERCC= sequencing quality control (consistency between replicates in the absence of biological variation and a capacity of 3'-end QuantSeq to quantify molar concentrations).

6. While the global decrease in miRNAs is interesting, one may argue that this is an artifact of the spike-in strategy. If the spike-ins aren't properly calibrated across each condition, the data will be mistakenly scaled up or down to give the illusion of a global increase or decrease. Additional controls and validation would be needed to support the conclusions made in the manuscript about miRNAs.

We understand that the way we presented the results could induce the reader to misinterpret our results. We would like to clarify that our goal was to normalize the RNA abundance by the cell numbers so we could achieve absolute quantification of the miRNAs on a per-cell basis. To that, we added the same volume (therefore same mass) of a commercially available small RNA spike-in mixture (Qiagen) to the cell lysates that contained the same number of neurons (10^6). The experiment was replicated two times to account for possible errors in cell number counts. We modified the figure legend to include that explanation. We also added a small schematic figure (new Fig 5A) to better describe the experimental design.

To further clarify the point, spike-in normalization error could be split into two classes, depending on its origin: random (counting cells or pipetting errors) and systematic (sequencing bias, for example, based on spike-in sample features). As in any experiment, the random component is reduced by doing replicates. While the systematic component is controlled by the diversity of spike-ins (molar concentration, sequence content).

To probe the random component, we reasoned that if there was a cell count or spike-in pipetting error in one of the samples, then the error bars for spike-ins would be larger than for endogenous miRNAs as this would be an additional source of error. We observed that spike-ins are measured as well as endogenous miRNAs (new Fig 5F). The goodness of the measurement was defined as a ratio between fold-change and its associated error bar size. This suggests the lack of a random component.

Next, we argue that the systematic component is also minimal because the commercial spike-in mixture, used in this study, contained ~50 different spike-in RNA molecules with a variety of sequences and molar concentrations. In addition, we observed that the measurement quality of spike-ins was similar across molar concentrations and G+C content (new Fig 5G).

Minor points:

1. Although differences across subcellular compartments in steady-state RNA-seq from the Boxer et al dataset imply the existence of post-transcriptional mechanisms affecting RNA dynamics in RTT, this is compatible with, but not equivalent, to the authors' findings about transcription rate and half-life. In another words, single-timepoint transcriptome profiling from the nucleus or from chromatin-associated RNA is not a sufficient proxy for transcription rate and single-timepoint whole-cell RNA-seq is not a sufficient proxy for the steady-state. The way this is discussed and the conclusions made in the text should be changed (e.g. lines 118-120, the labels in Figure 1F-H should name the different subcellular compartments, which are not the same as rates and steady-state. Furthermore, the analysis presented in the section titled "Half-life shifts and transcription buffering are conserved in *Mecp2* mouse models" should be changed to reflect differences in relative RNA abundance between genotypes across subcellular compartments – this is not directly equivalent to half-life changes and changes in transcription rate.

As suggested by the reviewer, we revised the labels in the identified figures to show the mouse subcellular compartment, and now refer to the compartments as "proxies" for transcription rate and steady-state in the text. We also refer to fold-change in mouse nuclear and whole-cell abundance rather than half-life and transcription rate in the identified section of text.

We agree that the Boxer et al mouse dataset is compatible with, but not directly equivalent, to our findings about transcription rate and half-life. Steady-state refers to a state, process, or quantity that does not change in time, and we use it in the restricted context of an equilibrium between transcription rate and half-life in the whole-cell only. In constant experimental conditions, this equilibrium from a single time point would not be expected to change over time.

In the nuclear subcellular fraction, we agree there is a distinction between transcription rate and mRNA abundance because there is a role for the nuclear export rate in determining the equilibrium mRNA abundance. However, the original Johnson et al and Boxer et al studies argue that the nuclear and chromatin fractions are indeed a good proxy of transcription rate in the context of *Mecp2* perturbations in mouse brain because:

- a) Nascent transcriptome measured by GRO-seq is similar to the nuclear fraction (Johnson et al. 2017).
- b) The nuclear and chromatin fraction data used in our study was shown to be a good proxy of transcription with alternative techniques like PRO-seq (Boxer et al. 2020).

On a separate note, the chromatin fraction is not affected by nuclear export and is expected to be closer to the transcription rate. Indeed, no mRNA is exported until it is detached from the chromatin. In summary, both papers support that the nuclear and chromatin data we used are good proxies of transcription rate and are compatible with our conclusions. Taken together, we believe that our mouse brain data re-interpretations are tracking closely both half-life and transcription rate changes found in the human neurons.

2. Although the authors have previously published the generation and characterization of their isogenic RTT iPSC-derived neurons, it is critical to include evidence showing the differentiation of human cortical neurons and the expression of MeCP2 between lines in the current study.

We thank the reviewer for noticing the lack of this critical piece of data. We added a western blot from protein lysates of the same neuronal lines showing the absence of MECP2 protein in the MECP2-null neurons (new Fig 1B). We also added a panel including the whole-cell mRNA abundances of known cortical neuronal marker genes in our derived neurons and how they compare to previously published in vitro-derived neurons using similar protocols and the fetal brain cortex at different gestational weeks.

3. In Figure 3, the source of the data displayed in panels F, G and H is unclear – is this a comparison between the *Mecp2*-null mice or the R306C mice, or were they merged? In Figure 3I these two lines are shown separately.

We apologize for the lack of information. We added critical information to the figures. In brief, Fig 3 G-I shows the data for the *Mecp2* y/- mice (*Mecp2*-null) and Fig 3J shows data for both *Mecp2* y/- and 306C genotypes.

4. Based on what is presented in Supplementary Figure 1, the 4sU-labeled fly spike-in RNA seems to be much more abundant than 4sU-labeled pulled-down human RNA. Will the excessive spike-in RNA lead to insufficient sequencing depth and inaccurate profiling? Dot blots would be more appropriate to show the input.

We understand how the referred figure can give that impression. However, the ribosomal band indicated for the fly RNA includes both 28S and 18S molecules. The summation of these ribosomal molecules gives the impression that there is more fly than human RNA. Moreover, our sequencing results show more human than fly reads as can be seen in figure S1E.

5. Critical information is missing from Figure 5 C and D. Do those boxplots show total fold-changes? Before and after scaling to spike-ins?

We added this information to the figure itself and figure legend.

6. Line 251-252: What about miRNA-binding sites in genes that only display changes in transcription rate and in genes that display buffering in the opposite direction (decreased TR and increased HL)?

We did not find any miRNA site enriched for the group of decreased TR and increased HL mRNAs, hence we omitted the figure. We modified the sentence in the results section to make clear that we did analyze this group of mRNAs however found no enrichment.

7. Supplementary Fig 1C, the genotype labels seem to be switched (the NEUwt should be over the right gel and NEUrtt should go over the left gel, given that the authors mentioned not being able to obtain the 1h timepoint in the RTT condition).

Thanks for spotting this. We re-labelled the figure.

REVIEWER COMMENTS

Reviewer #1 (Remarks to the Author):

Authors addressed the comments well and the manuscript is ready published.

Reviewer #2 (Remarks to the Author):

In this revision, the authors have addressed most of the concerns/questions raised in the first round of review. A few significant questions remain:

1. The RNA half-life calculation methodology remains problematic. Why isn't it possible to calculate half-lives with the saturation curve method for many lowly-expressed genes? The benefit of using the relative method is to have a large the number of genes quantified, but increasing the number of lowly-expressed genes may not be helpful since the calculations for those genes could be too noisy and dominate genome-wide averages.
2. The authors note that a pseudo-replicate was added to enhance the log₂ fold-change measurements, but this may artificially reduce variance and introduce bias. Including a pseudo-replicate is not recommended.
3. The transcription rates (TR) for WT and RTT samples are calculated differently, using multiple timepoints (WT) versus using one time point (RTT). The authors should calculate these values the same way to make genuine comparisons, just using one timepoint for each.
4. Code availability has yet to be addressed – did the authors check their Github link?

REVIEWER COMMENTS

Reviewer #1 (Remarks to the Author):

Authors addressed the comments well and the manuscript is ready published.

Reviewer #2 (Remarks to the Author):

In this revision, the authors have addressed most of the concerns/questions raised in the first round of review. A few significant questions remain:

1. The RNA half-life calculation methodology remains problematic. Why isn't it possible to calculate half-lives with the saturation curve method for many lowly-expressed genes? The benefit of using the relative method is to have a large the number of genes quantified, but increasing the number of lowly-expressed genes may not be helpful since the calculations for those genes could be too noisy and dominate genome-wide averages.

We understand how our ratio method strategy could lead readers to think the inclusion of low abundant or noisy genes may bias the conclusions. To address this, we added a new supplemental figure S3E (see below) showing that the buffering pattern remains the same when the analysis is done by including all genes (including low abundant genes, left panel), only genes with high transcription rate (TR) of >1 (top 70% of genes by transcription rate, middle panel) or only genes with even higher transcription rate of >10 (top 30% of genes by transcription rate, right panel). The presence of low abundant genes in our calculations does not induce the buffering observation as it is a common effect for genes regardless of their abundance levels.

Buffering is further supported by the high confidence mouse data using ten replicates for the WT and *MECP2*-null mutants that also captured the buffering effect using the proxy values for transcription rate. In this case, the large number of replicates eliminated the noise from the low abundance genes and we observed buffering in the absence of this potential noise in mice. In contrast, with the saturation curve method, the variance at each time point was prohibitively large to fit a satisfactory saturation curve for the low abundance genes.

2. The authors note that a pseudo-replicate was added to enhance the log₂ fold-change measurements, but this may artificially reduce variance and introduce bias. Including a pseudo-replicate is not recommended.

We now include a new Supplementary Figure 1P that reanalyzes the transcription rate calculations made with or without the pseudo-replicate in the context of our orthogonal validation of transcriptional rates using the EU incorporation assay followed by qRT-PCR (see below). We observed a minimal difference between calculations using both 0.5h and 1h samples (including the pseudo-replicate) compared with using just the 0.5h sample. Please note that the pattern does not change, indicating that the pseudo-replicate value does not bias measurements that have been experimentally validated, and that just the 0.5h sample is sufficient to mirror the experimentally validated measurements.

We agree that in future it is preferable to avoid the inclusion of a pseudo-replicate. Recognizing this, we adapted the Methods section describing the transcription rate calculation to highlight the potential pitfalls of using a pseudo-replicate. In brief, each time point of a RATEseq experiment is derived from a different well on the same multi-well plate. As a result, 0.5h & 1h samples account only for inter-well but not inter-plate variation and are susceptible to the same intra-plate bias.

3. The transcription rates (TR) for WT and RTT samples are calculated differently, using multiple timepoints (WT) versus using one time point (RTT). The authors should calculate these values the same way to make genuine comparisons, just using one timepoint for each.

We also recalculated the transcription rate shifts using only the 0.5h time point as a new Supplementary Figure 1K and compared it to the original analysis. We observed a correlation of 0.94 (both Pearson and Spearman). The scatterplot below compares the original and reanalyzed transcription rate shifts. Note the high similarity between the methods. Together with our response to point 2 above, these data argue that the measured transcription rate shifts from the original analysis are reliable.

4. Code availability has yet to be addressed – did the authors check their Github link?

We apologize for not ensuring the code was available during review and confirm that the Github link is active at <https://github.com/JellisLab/stabilome-rett>

REVIEWER COMMENTS

Reviewer #2 (Remarks to the Author):

In this revision, the authors have managed to have points 2-4 addressed, but need to re-calculate half-lives with the saturation curve method as requested the last time.

Reviewer #3 (Remarks to the Author):

The comments of Reviewer 2 addressed topics related to RNA half-life calculation, with special respect to low abundance mRNAs, pseudo-replicates and half-life fitting methods.

The study includes pseudo-replicates and not true biological replicates (comment 2).

Comment 1: It is difficult to assess the relation between RNA half-lives and RNA abundance because the RNA counts in WT and RTT settings were not provided (only the log₂FC values are given).

"Comment 3: The transcription rates (TR) for WT and RTT samples are calculated differently, using multiple timepoints (WT) versus using one time point (RTT). The authors should calculate these values the same way to make genuine comparisons, just using one timepoint for each."

While the authors show that the half-lives calculated by the above means are highly correlated (0.94), they do not show as to whether the comparisons and conclusions are affected or not. It would have been important to show that the RNA classes identified in this work do not change significantly. In other words, the authors do not provide a specific answer to the referee's comment, even though the high correlation of half-lives suggests that their findings are plausible. It is, however, possible that the genes shown in Fig. 2A are strongly affected by the relatively few uncorrelated mRNA half-lives.

RESPONSE TO REVIEWER COMMENTS

Reviewer #2 (Remarks to the Author):

In this revision, the authors have managed to have points 2-4 addressed, but need to re-calculate half-lives with the saturation curve method as requested the last time.

We have included the requested analysis as new figures S3G-H referred to on p8 of the text with an updated supplemental figure legend. We apologize for not addressing the request fully in the previous revision round. As can be seen, the half-life calculation using the saturation method does not alter the observation of the buffering effect. It is observed when including all or only the high-confidence quantified genes, demonstrating that the effect is not an artifact of lowly-expressed (noisy) genes.

Reviewer #3 (Remarks to the Author):

The comments of Reviewer 2 addressed topics related to RNA half-life calculation, with special respect to low abundance mRNAs, pseudo-replicates and half-life fitting methods.

The study includes pseudo-replicates and not true biological replicates (comment 2).

Comment 1: It is difficult to assess the relation between RNA half-lives and RNA abundance because the RNA counts in WT and RTT settings were not provided (only the log2FC values are given).

The raw counts for steady state RNA are summarized in figure 1D and for transcription rate in figure 1E. Importantly all the counts are available in the following 3 files:

GSE191168_counts_genes.csv.gz

GSE191168_counts_miRNA.csv.gz

GSE191168_counts_pA_sites.csv.gz

These files are available using the GEO access number: GSE191168, private access token: klmduakmjxivfkz. The data is currently private and can be accessed using the private access token. It will be made fully available upon manuscript acceptance.

"Comment 3: The transcription rates (TR) for WT and RTT samples are calculated differently, using multiple timepoints (WT) versus using one time point (RTT). The authors should calculate these values the same way to make genuine comparisons, just using one timepoint for each."

While the authors show that the half-lives calculated by the above means are highly correlated (0.94), they do not show as to whether the comparisons and conclusions are affected or not. It would have been important to show that the RNA classes identified in this work do not change significantly. In other words, the authors do not provide a specific answer to the referee's comment, even though the high correlation of half-lives suggests that their findings are plausible. It is, however, possible that the genes shown in Fig. 2A are strongly affected by the relatively few uncorrelated mRNA half-lives.

As requested, we now include a new figure S3B referred to on p8 of the text with an updated supplemental figure legend. The figure shows that the calculation using only the 0.5h time point on genes whose transcription rates were altered by at least 4-fold does not alter the main conclusion of the existence of a buffering effect nor the overall pattern of gene classes. It replicates the pattern of gene classes in the main figure 3E. These conclusions are corroborated by multiple independent analyses, including the measurement of transcription rates using the 5-EU method, the measurement of half-life using transcription inhibitor, and the re-analysis of the mouse datasets from the Greenberg lab.

REVIEWERS' COMMENTS

Reviewer #2 (Remarks to the Author):

The authors have addressed my question about mRNA hal-life calculation. My final comment is about the title and abstract, and I don't think the authors have addressed the conservative nature of their findings. The data from mice does not support half-life calculation as the authors claimed. On a minor note, the paper provided those data needs to be cited.

Reviewer #3 (Remarks to the Author):

The authors answered my comment only partially. They inserted Fig S3B and show a pattern of transcriptional and half-life changes in which the number of mRNAs in each class is similar with the two methods. However, most genetic or environmental perturbations do cause a shift in transcription rates and half-lives and similar overall patterns can easily arise. Thus, a pattern is not specific enough. To show that an RNA class does not change significantly, the authors should simply provide the following two items:

- 1) the list of mRNA names in each class common to both methods (in the form of an excel sheet).
- 2) the number of mRNAs in each class obtained with first, second and both methods. Specially, the table would look like:

Class	Multiple time points	Ratio-Method	Shared mRNAs
TR-only	299	266	195
Partial buffering....			

Only a sufficiently large number of shared mRNAs ensures that the two methods yield similar results.

REVIEWERS' COMMENTS

Reviewer #2:

The authors have addressed my question about mRNA hal-life calculation. My final comment is about the title and abstract, and I don't think the authors have addressed the conservative nature of their findings. The data from mice does not support half-life calculation as the authors claimed. On a minor note, the paper provided those data needs to be cited.

The conservation of buffering mechanism mentioned in the title, abstract, and throughout the text refers to the initial observation made in RTT mice by Johnson et al in 2017 (reference 17) that we demonstrate also exists in human RTT neurons. We refer to the initial observation in mice in the Abstract (line 22) and specifically to Johnson et al in the Introduction (lines 64 and 75). Our finding that a documented mouse mechanism is present in human RTT neurons on its own is enough to highlight that it is conserved.

Reviewer#2 presumably is referring to the mouse data from Boxer *et al.* (reference 9) that we use to orthogonally validate the changes in transcription rate and mRNA stability found in our datasets. Indeed, our reinterpretation of the mouse data from Boxer et al agrees with the Johnson et al results. They further strengthen our conclusion that the post-transcriptional regulation is conserved in mice and humans. We have already incorporated text changes to acknowledge that the Boxer et al results do not directly measure mRNA transcription or half-life but can be used as a proxy value for this. We estimated these changes utilizing their sequencing datasets of nuclear/chromatin-associated and whole-cell lysate RNAs. In summary, the papers providing the mouse data are extensively cited, and it is entirely justified to conclude as we do that the mechanism is conserved in RTT models.

Reviewer #3:

The authors answered my comment only partially. They inserted Fig S3B and show a pattern of transcriptional and half-life changes in which the number of mRNAs in each class is similar with the two methods. However, most genetic or environmental perturbations do cause a shift in transcription rates and half-lives and similar overall patterns can easily arise. Thus, a pattern is not specific enough. To show that an RNA class does not change significantly, the authors should simply provide the following two items:

1) the list of mRNA names in each class common to both methods (in the form of an excel sheet).
2) the number of mRNAs in each class obtained with first, second and both methods. Specially, the table would look like:

Class Multiple time points Ratio-Method Shared mRNAs

TR-only 299 266 195

Partial buffering....

Only a sufficiently large number of shared mRNAs ensures that the two methods yield similar results.

We thank the reviewer for this comment that in addition to the correlation we added to Supplementary Fig 3B last time, we should provide an excel table with the transcript names and their classification using both methods. We now provide a new **Supplementary Data table 4** precisely as requested, describing gene names and how they are consistent across mRNA half-life calculation methods.

We also provide a summary of these overlapping transcript numbers in a **new Supplementary Figure 3C**. This panel summarizes the similarity between class and gene identities, including a statistical demonstration of the high overlap between the tables of genes. In brief, we find that the percentage overlap ranges from 85-90% for the half-life only class, from 71-77% for the transcription rate only class, and from 34-60% for the full and partial buffering classes, with all Fisher p values exceeding 10^{-63} for every class. These calculations highlight the high statistical confidence in the specific gene overlaps regulated by post-transcriptional mechanisms found by both methods. The text in the main manuscript was modified to accommodate the new figures and table (main text lines 207-209).